# Selecting the most appropriate time points to profile in high-throughput studies

Michael Kleyman[1†], Emre Sefer[1†], Teodora Nicola[2], Celia Espinoza[3,4], Divya Chhabra[3,4], James S Hagood[3,4], Naftali Kaminski[5], Namasivayam Ambalavanan[2], Ziv Bar-Joseph[1*]

[1]Machine Learning and Computational Biology, School of Computer Science, Carnegie Mellon University, Pittsburgh, United States; [2]Division of Neonatology, Department of Pediatrics, University of Alabama at Birmingham, Birmingham, United States; [3]Division of Respiratory Medicine, Department of Pediatrics, University of California, San Diego, United States; [4]CARady Children's Hospital San Diego, San Diego, United States; [5]Section of Pulmonary, Critical Care and Sleep Medicine, School of Medicine, Yale University, New Haven, United States

**Abstract** Biological systems are increasingly being studied by high throughput profiling of molecular data over time. Determining the set of time points to sample in studies that profile several different types of molecular data is still challenging. Here we present the Time Point Selection (*TPS*) method that solves this combinatorial problem in a principled and practical way. *TPS* utilizes expression data from a small set of genes sampled at a high rate. As we show by applying *TPS* to study mouse lung development, the points selected by *TPS* can be used to reconstruct an accurate representation for the expression values of the non selected points. Further, even though the selection is only based on gene expression, these points are also appropriate for representing a much larger set of protein, miRNA and DNA methylation changes over time. TPS can thus serve as a key design strategy for high throughput time series experiments. Supporting Website: www.sb.cs.cmu.edu/TPS

*For correspondence: zivbj@cs.cmu.edu

[†]These authors contributed equally to this work

Competing interests: The authors declare that no competing interests exist.

## Introduction

Time series experiments are very commonly used to study a wide range of biological processes. Examples include various developmental processes (*Roy et al., 2010*), stem cell differentiation (*Sperger et al., 2003*), immune responses (*Yosef and Regev, 2011*), stress responses (*Gitter et al., 2013*) and several others. Indeed, analysis of the largest repository of gene expression experiments, the Gene Expression Omnibus (GEO), determined that roughly a third of these datasets come from experiments profiling dynamic processes over time (*Zinman et al., 2013*).

While mRNA gene expression data have been the primary source of high-throughput time series data, more recently several other genomic regulatory features are profiled over time. These include miRNA expression data (*Schulz et al., 2013*), ChIP-Seq studied to determine TF targets (*Chang et al., 2013*) and several types of epigenetic markers including DNA methylation (*Singer et al., 2014*), histone modifications (*Paige et al., 2012*) and more. In fact, with the rise in our ability to perform such high-throughput time series analysis, many researchers are now combining a few or several of these time series profiling experiments in a single experiment (*Chang et al., 2013*; *Buenrostro et al., 2015*) and then integrate these datasets to obtain a better understanding of cellular activity.

While integrated analysis of high-throughput genomic datasets can greatly improve our ability to model biological processes and systems, it comes at a cost. From the monetary point of view, these costs include the increased number of Seq experiments required to profile all types of genomic features. While such costs are common to all types of studies utilizing high-throughput data, they can be prohibitively high for time series based studies since they are multiplied by the number of time points required, the number of repeats performed for each time point and the number of different types of data being profiled. Importantly, even if the budget is not an issue, the ability to obtain enough samples for profiling all genomic features at all time points may be challenging, if not completely prohibitive.

One of the key determinants of the experimental and sample acquisition costs associated with time series studies is the number of time points that are being profiled. In most studies, the first and last time point can usually be determined by the researcher (for example, the time from birth to full lung structural development and maturation in mice). However, the number of samples required between these two points and the sampling frequency (given a fixed budget) are often hard to determine based on phenotypic observations since the molecular events of interest may precede such phenotypic events. To date, sampling rates have largely been determined using one of two ad-hoc protocols. The first utilized uniform sampling across the duration of the study (*Li et al., 2013*) with the number of samples constrained by the available budget and samples. The second relied on some (conceived or real) knowledge of the process, often based on phenotypic observations. These studies, especially for responses though also for development, have often used nonuniform sampling (*Schulz et al., 2013*; *Bar-Joseph et al., 2003a*) though it is hard to determine if such sampling misses important molecular events between the sampled points.

Relatively, little work has focused so far on the selection of time points to sample in high through-put time series studies. Singh et al (*Singh et al., 2005*) and Rosa et al (*Rosa et al., 2012*) presented an iterative process which starts with profiling a small number of time points and then selects the next time point either based on an Active Learning method (*Singh et al., 2005*) or based on using prior related experiments (*Rosa et al., 2012*). Next the selected point is profiled and the process is repeated until a stopping criteria has been reached. Both of these methods require several iterations until the final time series is profiled, which can drastically lengthen the experiment time and can introduce additional biases making them less useful in practice. In addition, these methods employ a stopping criteria that does not take into account the full profile and also require that related time series expression experiments be used to select the point, which may be problematic when studying new processes or treatments.

Here, we propose the first non iterative method to address the issue of sampling rates across all different genomic data types. Our method starts by selecting a small set of genes that are known to be associated with the process being studied (while the full set is often unknown, for most processes a small set is usually known in advance). Next, we use a cheap array-based technology to sample these genes at a high, uniform rate across the duration of the study. Note that unlike standard curve fitting algorithms, a method for selecting time points for these experiments is required to accommodate over a hundred curves (for all genes) simultaneously, and we discuss various ways to formulate this as an optimization problem. To solve this optimization problems, we developed the Time Points Selection method (*TPS*), an algorithm that uses spline based analysis and combinatorial search to select a subset of the points that, when combined, provide enough information for reconstructing the values for all genes across all time points. The number of points selected can either be set in advance by the user (for example, based on budget constraints) or can be defined as a function of the reconstruction error. The selected time points are then used for the larger, genome-wide experiments across the different types of data being profiled.

To test and evaluate the method we applied it to study lung development in mice. Normal development of lung alveoli through the process of alveolar septation is a dynamic, coordinated process that requires the accurate spatial and temporal integration of signals. We currently lack a comprehensive understanding of the dynamic networks that govern normal alveolar septation. Thus, lung development can serve as an ideal test case for TPS since a variety of different time series genomic datasets are needed to enable accurate reconstruction of networks regulating this process. As we show, *TPS* was able to successfully identify time points for reconstructing the mRNA profiles of selected genes and these points improved upon uniform based sampling for such points. Further, we show that the set of points selected based on the analysis of this limited set of highly sampled

mRNAs is also appropriate for sampling a much larger, unbiased, set of miRNA profiles as well as to determine the temporal protein levels of over 1000 proteins. Finally, we show that the mRNA samples can also be used to determine the optimal sampling points for a DNA methylation study of the same developmental process.

## Results

### The time points selection (*TPS*) method

We developed *TPS* to select a subset of $k$ time points from an initial larger set of $n$ points such that the selected subset provides an accurate, yet compact, representation of the temporal trajectory. *Figure 1* presents an overview of the method. *TPS* utilizes splines to represent temporal profiles and implements a cross-validation strategy to evaluate potential sets of points. Following initialization which is based on the expression values, we employ a greedy search procedure that adds and removes points until a local minima is reached (Materials and methods). The resulting set is then used for the larger genomic and epigenetic experiments.

To test the usefulness of *TPS*, we used it to determine time points for a lung development study in mice. We first profiled the expression of 126 genes known or suspected to be involved in lung development using NanoString (See Appendix Methods for a list of the selected genes and the reason each was selected). We then used *TPS* analysis of these experiments to select a subset of time points for profiling the expression of a larger, unbiased, set of miRNAs. Finally, we have used *TPS* to design time series experiments to study DNA methylation patterns for a subset of the genes.

### *TPS* identifies subset of important time points across multiple genes

We have tested the performance of *TPS* by using it to select subsets of points ranging from 3 to 25 and evaluating how well these can be used to determine the values of non-sampled points. To determine the accuracy of the reconstructed profiles using the selected points, we computed the average mean squared error for points that were not used by the method (Materials and methods). The results are presented in *Figure 2*. The figure includes a comparison of our method with two baseline methods: a random selection of the same number of points and uniform sampling of points within the range being studied, a method that is commonly used for time series expression profiling as discussed above. We have also compared the performance of the different strategies for initializing the set of points as discussed in Appendix Method (sorting by absolute differences or by equal partition) and between different methods for searching for the optimal subset (simulated annealing, weighting genes by cluster size, and adding/removing multiple time points per iteration, Appendix Methods). Finally, *Figure 2* also presents the repeat noise values which is the theoretical limit for the performance of any profile reconstruction method.

As expected, we find significant performance improvement when using *TPS* when compared to randomly selected points. Importantly, we also see a significant and consistent improvement (for all sizes of selected time points) over uniform sampling highlighting the advantage of condition-specific sampling decisions. Sorting initial points by absolute values further improves the performance highlighting the importance of initialization when searching large combinatorial spaces. Simulated annealing, weighting, and multiple point selection improve performance as well. As the number of points used by *TPS* increases, it leads to results that are very close to the error represented by noise in the data (0.108) ( *Figure 2—figure supplement 1*).

*Figure 3* presents the reconstructed and measured expression values when using *TPS* to select 13 time points (less than a third of the points that were profiled). Note that even though each of these genes has distinct trajectory and inflection points, the selected set of time points enable *TPS* to fit all quite accurately without overfitting (See *Figure 3—figure supplement 1* and *Figure 3—figure supplement 2* for figures of several other genes and for figures reconstructed by using the best 8 time points as determined by *TPS*, respectively).

### Identified time points using mRNA data are appropriate for miRNA profiling

To test the usefulness of our method for predicting the correct sampling rates for other genomic datasets, we next profiled mouse miRNAs for the same developmental process. miRNAs have been

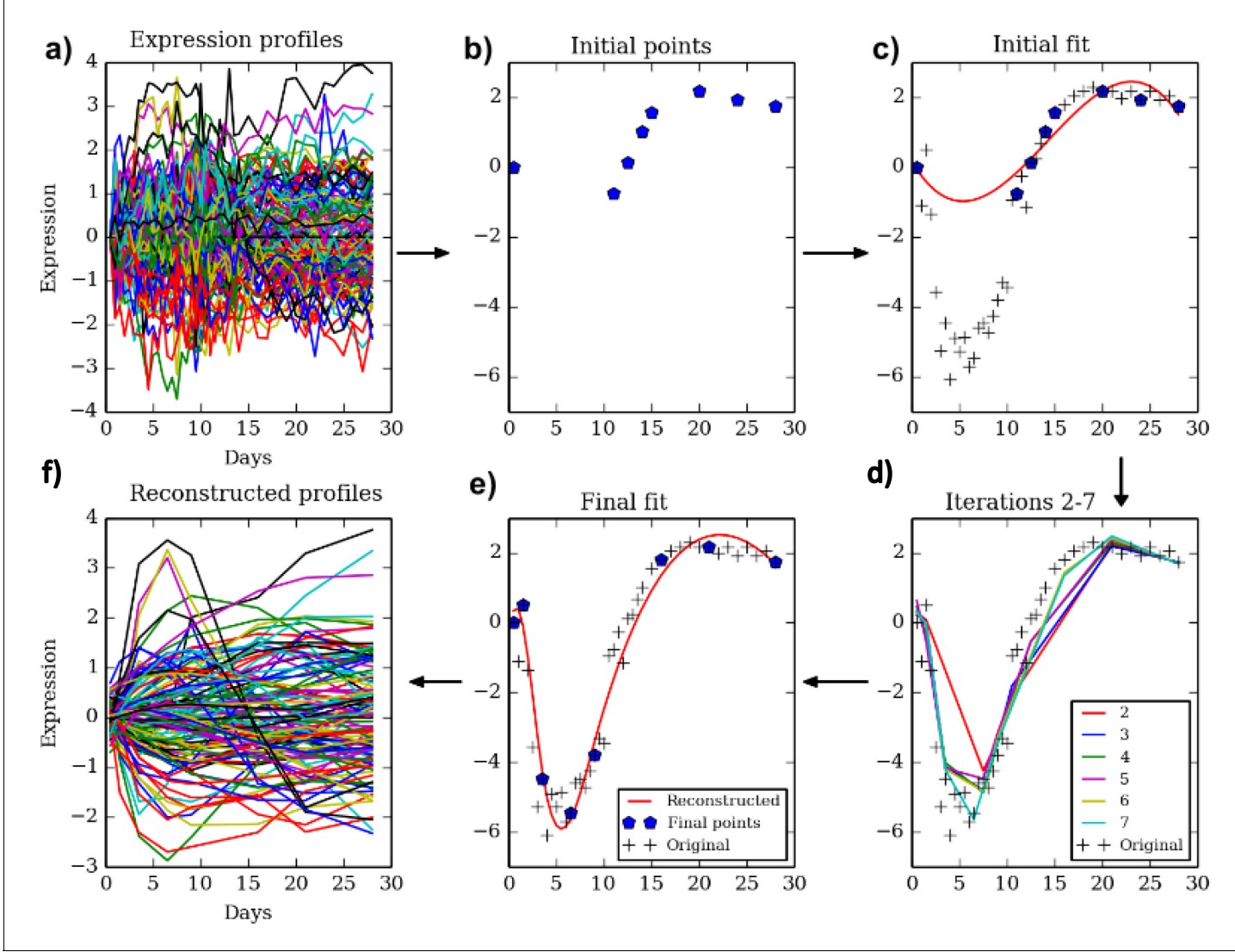

**Figure 1.** The TPS method. Clockwise from top left. Given a dense sampling of a selected subset of genes (**a**) we select an initial set of points (**b**) using the initialization method described in the text. Next, we fit a spline to the selected points for each gene (**c**) and evaluate the error on all other points. We perform a greedy search process (**d**) which iteratively removes and adds points to improve the test data fit resulting in the final set of points (**e**). The reconstructed curves are fitted to all genes (**f**) and an overall error is computed and compared to the theoretical limit (noise) to determine the ability of the selected number of points to fit the data.

The following figure supplements are available for figure 1:

**Figure supplement 1.** Comparison of performance between TPSand a previous method Singh et al.

**Figure supplement 2.** Comparison of initialization methods to each other by their final error.

**Figure supplement 3.** Comparison of initialization method by their final error compared to selecting random points.

known to regulate lung development (*Sessa and Hata, 2013*) and several miRNAs are differentially expressed during this developmental process (*Williams et al., 2007*). Several of these are also coordinately activated with various TFs to control specific transitions during development (*Schulz et al., 2013*). Thus, any large scale effort to model lung development would require the profiling of miRNAs as well. Unlike the mRNA dataset, which utilized prior knowledge to profile less than 1% of all

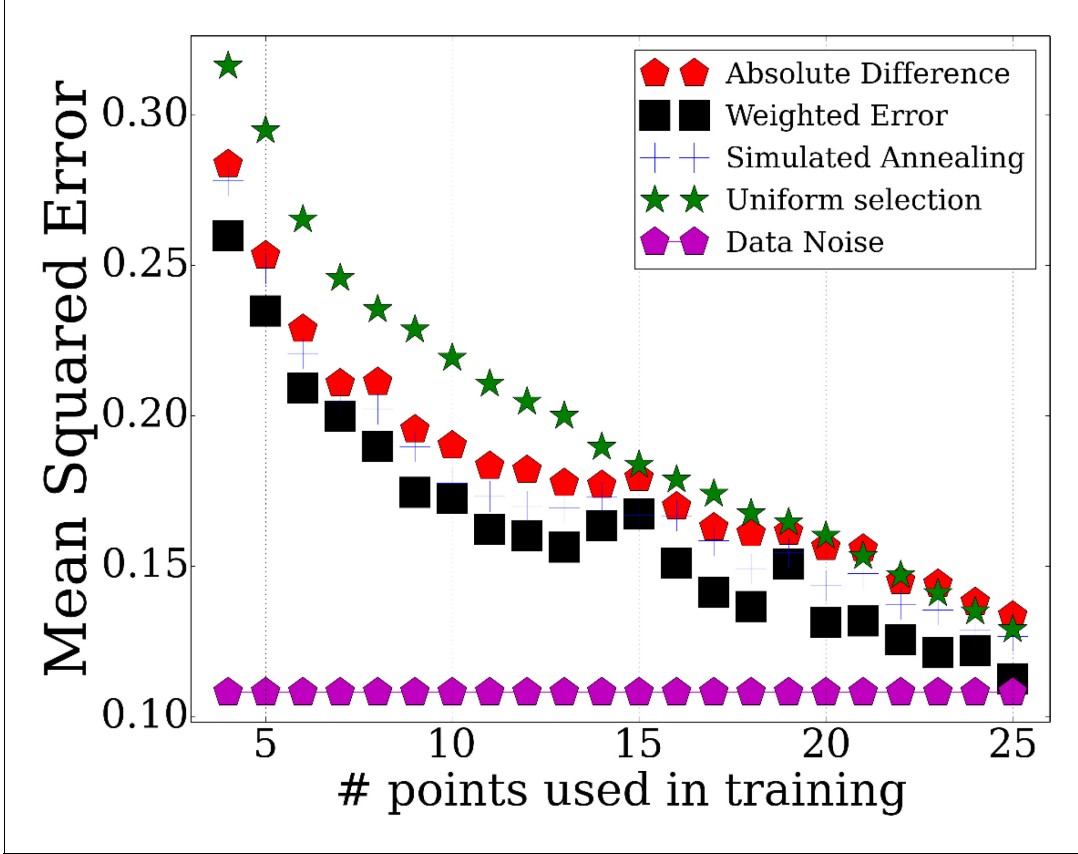

**Figure 2.** Performance of TPS using different sizes for the selected points. Error comparisons of TPS variants to uniform selection of points and noise. Absolute difference - Greedy iterative addition with absolute difference initialization (Algorithm 1, Appendix Methods). Simulated annealing - Iterating using simulated annealing with absolute difference initialization. Weighted error - Selection based on cluster rather than individual gene errors. See Appendix Methods for details.

The following figure supplements are available for figure 2:

**Figure supplement 1.** Average noise in each mRNA expression time point.

**Figure supplement 2.** Comparison of error for the TPS algorithm on full data, 75% random data, and random points chosen on the full data.

**Figure supplement 3.** Comparison of TPS and piecewise linear fitting over genes (a) *Pdgfra*, (b)*Eln*, (c) *Lrat*.

**Figure supplement 4.** Comparison of the reconstruction error when using the points selected by TPS and when using the same number of random points from the overall set of sampled points.

genes, the miRNA dataset contained a much larger number of miRNAs ($\hat{6}00$). Thus, the miRNA data represent an unbiased sample providing information on whether using one type of genomic data can be helpful for determining rates for other types. In our analysis, we normalized miRNA values by variance mean normalization (***Bolstad et al., 2003***).

To test *TPS* on this dataset, we used the *mRNA* expression data to select time points and then used the miRNA expression values for the selected time points to reconstruct the complete trajectories for each miRNA. The results are presented in ***Figure 4***. As can be seen, when using the points selected based on the mRNA data we achieve a much lower error when compared to the error resulting from using the same number of uniform or random points ($p<0.01$ for random based on randomization analysis) highlighting the relationship between the two datasets and the ability to use one to determine points for the other. More generally, even though the noise in the miRNA data is

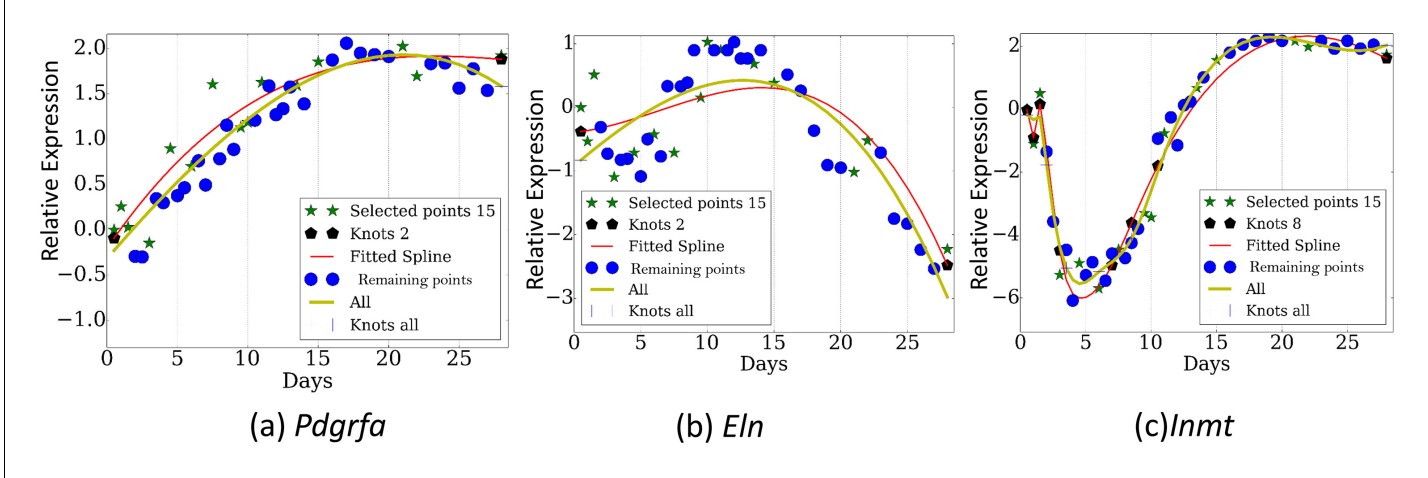

(a) *Pdgrfa*  (b) *Eln*  (c)*Inmt*

**Figure 3.** Reconstructed expression profiles for selected genes. (**a**). Pdgfra. , (**b**). Eln. , (**c**). Inmt.

The following figure supplements are available for figure 3:

**Figure supplement 1.** Expression profiles over several genes (**a**) *Esr2*, (**b**) *Nme3*, (**c**) *Polr2a*.

**Figure supplement 2.** Reconstructed expression proles by eight points over genes (**a**) *Pdgfra*, (**b**) *Eln*, (**c**) *Inmt*.

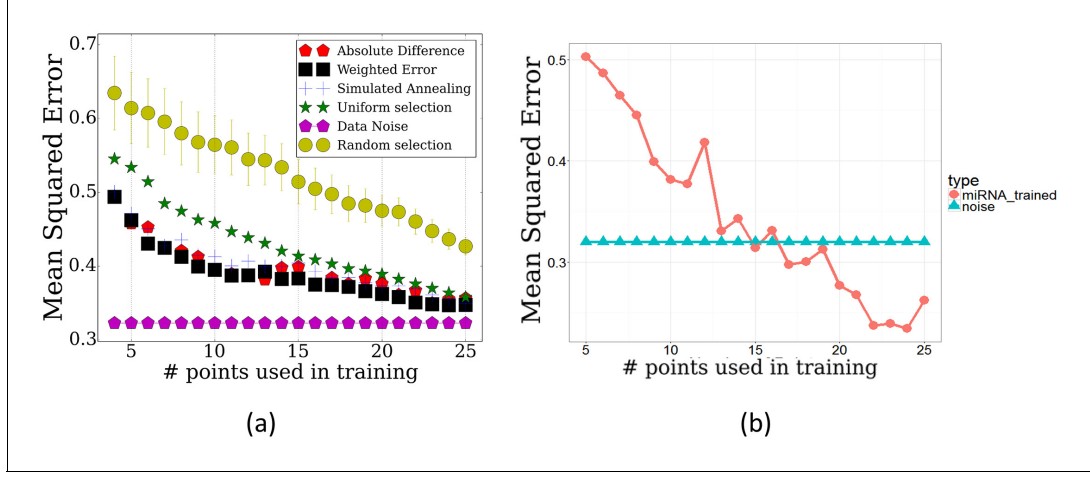

(a)                                      (b)

**Figure 4.** Performance of TPS by on the miRNA data. (**a**) TPS reconstruction error when using the mRNA data to select time points for the miRNA experiments. Results of random and uniform selection as well as repeat noise error are also presented for comparison. TPS variants shown are the same two presented in *Figure 2*. (**b**) Error of splines with points selected by training TPS on the actual miRNA data itself, using the maximum absolute difference initialization.

The following figure supplements are available for figure 4:

**Figure supplement 1.** Observed and reconstructed expression proles for miRNAs (**a**) *mmu-miR-100*, (**b**) *mmu-miR-136*,c) *mmu-miR-152*, (**d**) *mmu-miR-219*.

**Figure supplement 2.** 8 stable miRNA clusters.

**Figure supplement 3.** TPS performance for the proteomics data using different number of time points.

higher than for the mRNA dataset, relative ordering of the performance of each of the methods is similar to the mRNA results in *Figure 2*. This serves as a strong indication that mRNAs can serve as a general proxy for selecting time points for other genomic datasets. *Figure 4b* presents the error achieved when using the miRNA data itself to select the set of points (evaluated on the miRNA data). As expected, the performance when using the miRNA data itself is better than when using the mRNA data. However, when taking into account the inherent noise in the data the differences are not large. For example, when using the 13 selected mRNA points, the average mean squared error is 0.4312 whereas when using the optimal points based on the miRNA data itself the error is 0.4042.

   *Figure 4—figure supplement 1* presents the reconstructed and measured expression values for a few miRNAs based on time points identified using the mRNA dataset. As with the mRNA data, the ability to accurately reconstruct different miRNA profiles highlights the importance of selecting a global set of points that can fit all genes and miRNAs in our study.

   We have also analyzed the performance of *TPS* when using the mRNA data to select sampling time points for profiling the levels of more than 1000 proteins. We observed results that are very similar to the results obtained for the miRNA time point selection. Specifically, the points selected by TPS lead to reconstruction errors that are lower than those observed for uniform sampling or for a random set of the same number of points further demonstrating the general applicability of our method. See Appendix Results for details.

## Using *TPS* to select time points for DNA methylation analysis

In addition to mRNA and miRNA expression data, epigenetic data have been increasingly studied in time series experiments (*Talens et al., 2010*; *Schneider et al., 2010*). To test the ability of the mRNA data to determine the appropriate points for DNA methylation analysis, we used targeted bisulfite sequencing to profile three CpG-enriched regions for 13 genes at 8 of the 42 time points used for the mRNA and miRNA studies (Materials and methods). We next applied TPS to the mRNA data of these 8 points to select the best subset of 4 points and compared the selected points to those that would have been selected using the methylation data itself. The 4 points identified using the mRNA data (0.5, 5, 15, 26) were exactly the same as the ones selected using the methylation data indicating again that mRNA data is a good proxy for the dynamics of the epigenetic data as well. *Figure 5—figure supplement 1* presents the reconstructed splines over the identified points for several genomic methylation loci. *Figure 5* presents the methylation and expression curves for 3 genes: *Akt,1 Cdh11*, and *Tnc*. These were the genes with the strongest negative correlation between their methylation and expression. As can be seen, in several cases we observed strong negative or positive correlations between the two datasets in the time points we used serving as another indication for the ability to use one dataset to select the sampling points for the other. See *Figure 5—figure supplement 2* for correlation of all genes.

## Discussion

Time series gene expression experiments are widely used in several studies. More recently, advances in sequencing and proteomics are enabling the profiling of several other types of genomic data over time. Here we focused on lung development in mice with the goal of identifying an optimal set of time points for profiling various genomic and proteomic data types for this process.

   An important question is: Whether a better selection of time points really leads to observations that are missed when using an inferior set of points (even if the number of points is the same)? To answer this question we looked at several prior studies that profiled mouse lung development over time using various high throughput assays. *Table 1* presents 9 representative studies and lists the biological data that was profiled and the time points that were used. As can be seen, while certain time points seem to be widely used across studies (for example, 7d) others were profiled in only one or two of the studies (2d, 10d, three weeks). This raises several issues. First, it is very hard to compare or combine these datasets (for example, protein levels were not profiled on day 7(*Cox et al., 2007*) whereas all mRNA levels were). It also makes it hard to determine if differences between DE genes or miRNAs between these studies are the result of differences in the underlying conditions studied (for example, when testing for mutants or treatments) or simply the result of different sampling. Finally, each of these studies may have missed key genes, proteins or miRNAs because of the

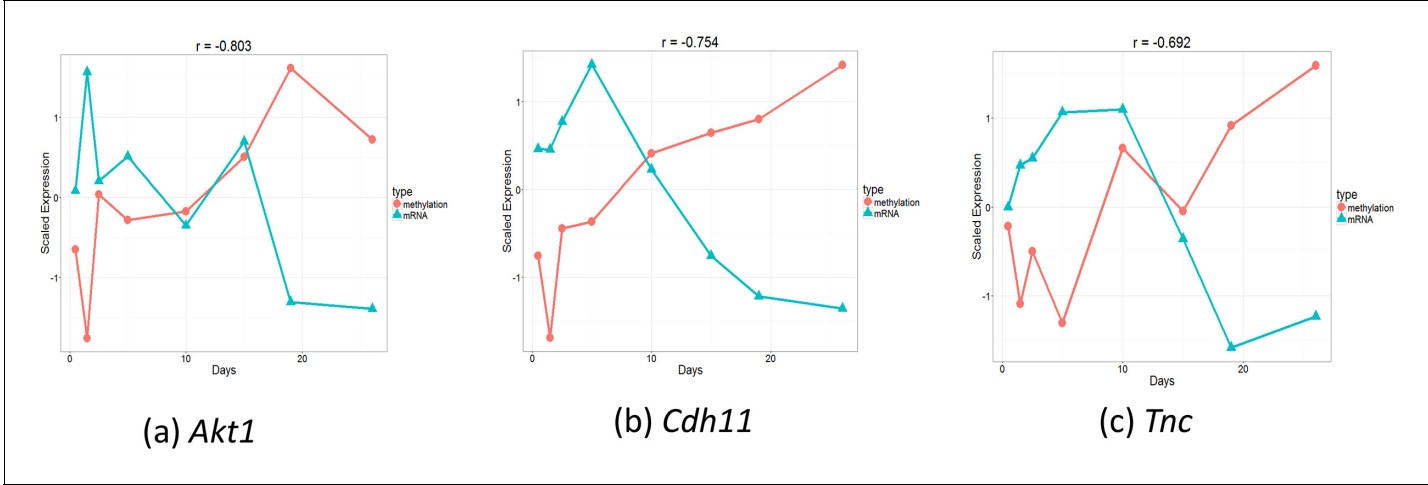

**Figure 5.** Comparison of gene expression and methylation data for selected genes. (**a**). Akt1. , (**b**). Cdh11. , (**c**). Tnc.
The following figure supplements are available for figure 5:

**Figure supplement 1.** Reconstructed methylation proles over several loci (chromosome, position) with corresponding genes.
**Figure supplement 2.** Bootstrap analysis of Pearson correlation r between expression and methylation datasets over eight time points for each gene.

sampling used restricting the ability of downstream analysis to use the data to model causal and regulatory events in lung development.

To illustrate these problems we compared the resulting curves using three of the sampling rates from *Table 1* to the reconstructed curves obtained by using TPS to select the optimal 5 and 8 time points. For example, the points selected by *Schulz et al. (2013)* are 0, 4, 7, 14 and 28 (since 28 is last day in our analysis we used it instead of 42). In contrast, TPS selects 0.5, 6, 9.5, 19 and 28. As can be seen in *Figure 6*, important expression changes in key genes are missed by using the arbitrary points while the TPS points are able to correctly reconstruct these profiles even though the total number of points is the same (5). More globally, the error for the arbitrary set of selected points is much higher on average (*Appendix 2—Table 4*). Similar results are obtained for the other sampling rates used in the past (*Figure 6*, *Appendix 2—Table 4*) and when comparing TPS to iterative methods previously suggested for selecting the set of points to profile (*Figure 1—figure supplement 1*). This indicates that accurate selection of time points can have a large impact on the ability of the study to identify

**Table 1.** Summary of prior high throughput lung development studies.

| Reference | Data types | Selected time points (Days) |
| --- | --- | --- |
| [*Bonner et al., 2003*] | mRNA expression | E9, E4, E17, 0, 7, 14, 28 |
| [*Melén et al., 2011*] | mRNA expression | E16, E18, 0, 7, 14, 28 |
| [*Bhaskaran et al., 2009*] | microRNA expression | E16, E19, E21, 0, 6, 14, 60 |
| [*Dong et al., 2011*] | mRNA and microRNA expression | E12, E14, E16, 0, 2, 10 |
| [*Cox et al., 2007*] | Protein expression levels | E12, E14, E18, 2, 14, 56 |
| [*Schulz et al., 2013*] | mRNA and miRNA expression | 0, 4, 7, 14, 42 |
| [*Cormack et al., 2010*] | mRNA expression | 0, 7, 14, adult |
| [*Mager et al., 2007*] | mRNA expression | E15, E17, E19, E21, 1, 14, 84 |
| [*Mariani et al., 2002*] | mRNA expression | E18, 1, 4, 7, 10, 14, 21, adult |

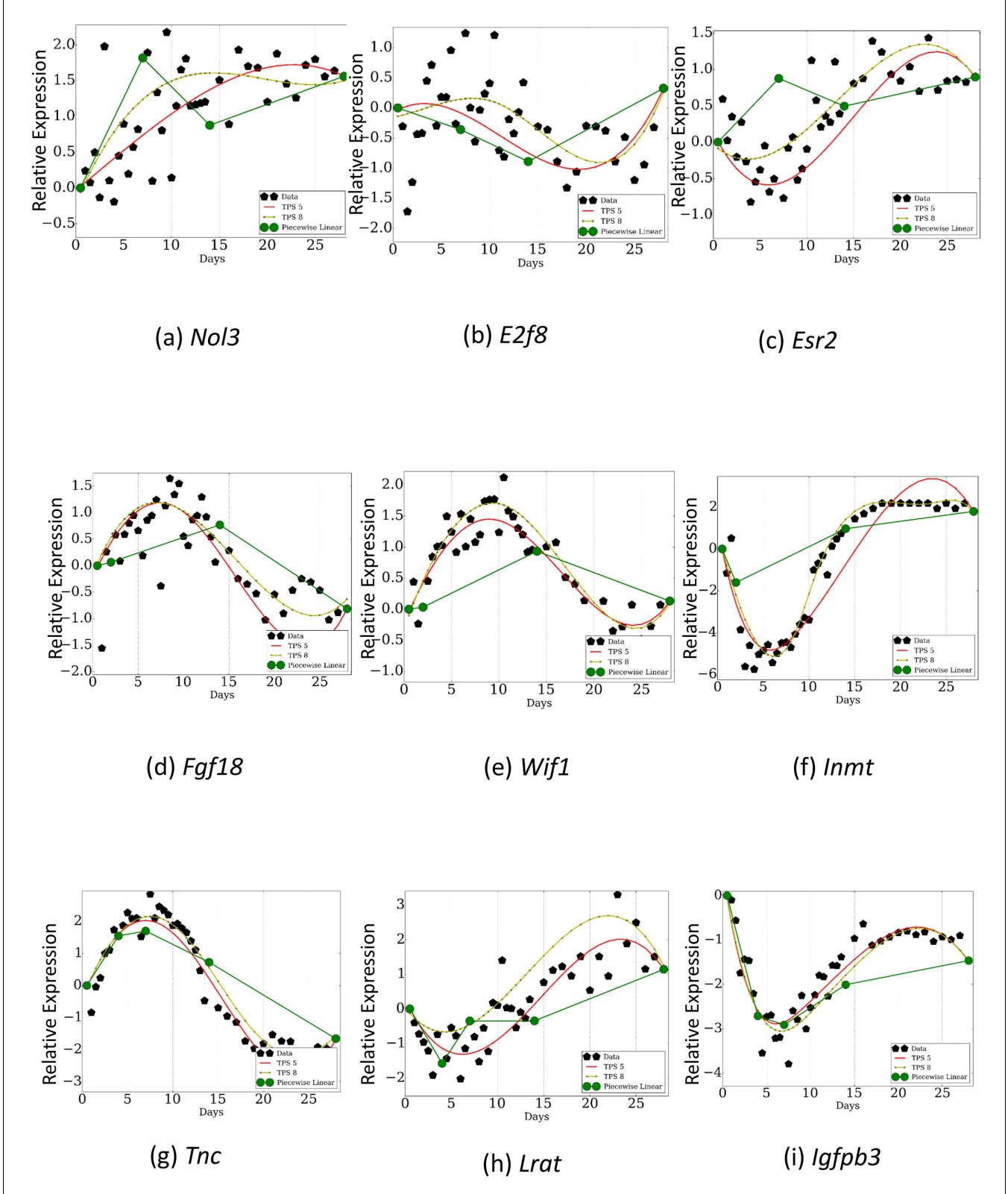

**Figure 6.** Comparison of TPS with sampling rates used in previous studies. Dark green curves are the reconstructed profiles based on the points profiled by prior studies. Light green and red curves are based on the points selected by TPS . As can be seen, even when comparing results from using the same number of points, TPS can identify key events for some of the genes that are missed when using the phenotype based sampling rates. *Figure 6 continued on next page*

*Figure 6 continued*

Subfigures a,b, and c are a piecewise linear fit over points 0.5, 7.0, 14.0, 28.0 . Subfigures d,e, and f are a piecewise linear fit over points 0.5, 2.0, 14.0, 28.0. Subfigures g,h, and i are a piecewise linear fit over points 0.5, 4.0, 7.0, 14.0, 28.0.

The following figure supplement is available for figure 6:

**Figure supplement 1.** Comparison of gene expression and protein abundance for selected gene protein pairs.

key genes and events. See also Appendix Results for a discussion about the importance of the differences between the TPS and prior work results for selected genes.

Our method relies on a very small subset of genes that are known to be involved in the process studied for the initial (highly sampled) set of experiments. While such set is known for several processes, there may be cases where very little is known about the biological process and so it may be hard to obtain such set. TPS can still be applied to determine sampling rates for such processes using a small *random* set of genes. To illustrate this we repeated the analysis presented in Results using only the measured values of 25% of genes in our original set and replacing the values for the other genes with random profiles. As we show in *Figure 2—figure supplement 2*, even when using such set, the time points selected by TPS greatly improve upon an arbitrary set of the same number of time points. Since in most time series experiments at least 25% of the genes are differentially expressed (and in several cases a much larger fraction, (*Zhou et al., 2009*; *Shi et al., 2015*) a random selection of genes is likely to exhibit similar results even for poorly understood processes.

Beyond the analysis of a specific type of data, several studies have now been profiling multiple types of genomic data over time. Such studies need to agree on a set of time points which would be common to all experiments so that these diverse types can be integrated to form a unified model (*Chang et al., 2013*; *Roy et al., 2010*). To date, the selection of such points relied on ad-hoc methods. The processes being studied were either sampled uniformly or based on prior knowledge. However, known properties of such systems were often been based on phenotypic observations which may not necessarily agree with the timing of molecular events. In addition, in many case studies of the same, or similar processes differed with respect to the time points that have been profiled. For example, early work on the analysis of cell cycle data in yeast utilized both uniform and nonuniform sampling (*Spellman et al., 1998*) and recent studies of circadian rhythms have followed a similar pattern (*Storch et al., 2002*; *Ueda et al., 2002*). Similarly, more recent analysis of responses to flu diverged widely in the (nonuniform) sampling rates that were used (*Shapira et al., 2009*; *Li et al., 2011*).

*TPS* addresses these problems by using a principled method for determining sampling rates. An important goal in the development of *TPS* was to enable it to be successfully applied to different types of biological datasets. As we show, a relatively inexpensive, gene centric, method provides a very good solution for RNA expression profiling as well as other types of data including miRNAs and DNA methylation. Thus, a combined experiment can be fully designed using our method.

While we evaluated TPS on several types of high throughput data, we have only tested it so far on data for a specific biological process (lung development in mice). While we believe that such data is both challenging and representative and thus provides a good test case for the method, analysis of additional datasets may identify new challenges that we have not addressed and we leave it to future work to address these.

TPS, including all initialization methods discussed, is implemented in Python and is available on the supporting website. We hope that as sequencing technology continues to advance, more and more studies would integrate diverse types of time series data and will utilize *TPS* in the design pipeline of their studies.

## Materials and methods

### mRNA and miRNA used in the study

To select the list of 126 genes used in the NanoString profiling we searched the literature for genes that have been linked to the following processes: (a) Cell type specification genes (e.g. alveolar type

I epithelial, alveolar type II epithelial, any epithelial, basal, endothelial, mesenchymal, pericyte, fibroblast, monocyte), (b) genes known to be up or down regulated during septation, (c) genes known to be altered in DNA methylation during development, (d) genes known to be involved in septation, (e) genes known to be regulated by miRNA involved in septation, and (f) genes known to be regulated by DNA methylation during fibrosis. *Appendix 2—Table 1* contains a list of the selected genes and the process for which they were selected.

For the miRNA set we used a commercially available, unbiased, array (the nCounter Mouse miRNA Expression Assay Kit, NanoString).

## mRNA and miRNA profiling and analysis

A total of 240 samples were isolated by Laser Capture Microscopy (LCM) from murine lung at multiple time points (E16.5, P.05 to P14 every 12 hr, and P15 to P28 every 24 hr). The samples were used to prepare total RNA. RNA extraction was performed by miRNeasy MicroKit (Qiagen) following the manufacturer's protocol. RNA concentration and integrity were measured by using NanoDrop ND-2000 and 2200 Tape Station. A custom NanoString probe set (Reporter Code set and Capture Probe set) for 126 genes was designed and the nCounter Gene Expression Assay was performed using 50 ng total RNA. The data files produced by the nCounter Digital Analyzer were exported as a Reporter Code Count (RCC) file and data normalization was performed using the nSolver, the analysis software provided by Nanostring.

## DNA methylation analysis

Mouse alveolar lung tissues attached to LCM caps were stored at −80°C until processing. DNA was extracted using the ZR Genomic DNA-Tissue MicroPrep kit (Zymo Research). Incubation with Digestion buffer and proteinase K was done overnight at 55°C in inverted tubes. 13 genes were chosen for targeted NextGen bisulfite sequencing (NGBS): *Igfbp3, Wif1, Cdh11, Eln, Sox9, Tnc, Dnmt3a, Akt, Vegfa, Lox, Foxf2, Zfp536* and *Src*, based on published data (*Cuna et al., 2015*). Targeted NGBS was done on samples collected at: E16.5, E18.5, P0.5, P1.5, P2.5, P5, P10, P15, P19 and P26. Multiplex PCR was performed using 0.5 units of TaKaRa EpiTaq HS (Takara Bio, Kusatsu, Japan) in 2x master mix. FASTQ files were aligned using open source Bismark Bisulfite Read Mapper using Bowtie2. Methylation levels were calculated in Bismark. Sites where the difference in methylation was less than 5% over the entire time period, those where there was a difference of >20% at a single time point and those with less than 3 non zero values were removed from the analyses.

## Problem statement

Our goal is to identify a (small) subset of time points that can be used to accurately reconstruct the expression trajectory for *all* genes or other molecules being profiled. We assume that we can efficiently and cheaply obtain a dense sample for the expression of a very small subset of representative genes (here we use nanostring to profile less than 0.5% of all genes) and attempt to use this subset to determine optimal sampling points for the entire set of genes.

Formally, let $G$ be the set of genes we have profiled in our dense sample, $T = \{t_1, t_2, \ldots, t_T\}$ be the set of all sampled time points. We assume that for each time point we have $R$ repeats for all genes. We denote by $e_{gt}^r$ be the expression value for gene $g \in G$ at time $t \in T$ in the $r$'th repeat for that time point. We define $D_g = \{e_{gt}^r, t \in T, r \in R\}$ as the complete data for gene $g$ over all replicates and time points $T$.

To constrain the set of points we select, we assume that we have a predefined budget $k$ for the maximum number of time points we can sample in the complete experiment (i.e. for profiling all genes, miRNAs, epigenetic marks etc. using high-throughput seq experiments). We are interested in selecting $k$ time points from $T$ which, when using only the data collected at these $k$ points, minimizes the prediction error for the expression values of the unused points. To evaluate such a selection, we use the selected values to obtain a smoothing spline (*De Boor, 1978*; *Bar-Joseph et al., 2003a*; *Wahba, 1990*) function for each gene and compare the predicted values based on the spline to the measured value for the non-selected points to determine the error. In our problem, $t_1$ and $t_T$ define the first and end points, so they are always selected. The rest of the points are selected to maximize the following objective 1:

Problem statement: *Given $D_g$ for genes $g \in G$, the number of desired time points $k$, identify a subset of $k - 2$ time points in $T \setminus \{t_1, t_T\}$ which minimizes the prediction error for the expression values of all genes in the remaining time points.*

## Spline assignments

Before discussing the actual procedure we use to select the set of time points, we discuss the method we use to assign splines based on a selected subset of points for each gene. There are two issues that need to be resolved when assigning such smoothing splines: (1) The number of knots (control points) and (2) their spacing. Past approaches for using splines to model time series gene expression data have usually used the same number of control points for all genes regardless of their trajectories (*Subhani et al., 2010*; *Bar-Joseph et al., 2003b*), and mostly employed uniform knot placements. However, since our method needs to be able to adapt to any size of $k$ as defined above, we also attempt to select the number of knots and their spacing. We do this by using a regularization parameter for the fitted cubic smoothing spline where number of knots is increased until the smoothing condition is satisfied (*Wahba, 1990*). The regularization parameter is estimated by leave-one-out cross-validation (LOOCV).

## *TPS* : Iterative process to select points

Because of the highly combinatorial nature of the time points, we rely on a greedy iterative process to select the optimal points as summarized in *Figure 1* (See Appendix Methods for pseudocode).

There are three key steps in this algorithm which we discuss in detail below.

- *Selecting the initial set of points:* When using an iterative algorithm to solve non-convex problems with several local minima, a key issue is the appropriate selection of the initial solution set (*Hartigan, 1975*; *McLachlan and Peel, 2004*)]. We have tested a number of methods for performing such initializations and results for some of these are presented in *Figure 1—figure supplement 2*. Since the goal of the method is to optimize a specific function (error on the left out set of expression values measured at time points not used), all initialization methods can be tested for each dataset and the solution minimizing the left out error can be used. See Appendix Methods for details.

- *Iterative improvement step:* After selecting the initial set, we begin the iterative process of refining the subset of selected points. In this step we repeat the following analysis in each iteration. We exhaustively remove all points from the existing solution (one at a time) and replace it with all points that were not in the selected set (again, one at a time). For each pair of such point, we compute the error resulting from the change (using the splines computed based on the current set of points evaluated on the left out time points), and determine if the new point reduces the error or not. Formally, let $T^- = T \setminus \{t_1, t_T\}$ and $C_n$ be set of points for iteration $n$. We are interested in finding a point pair $(t_a \in C_n, t_b \in T^- \setminus C_n)$ which minimizes the following error ratio for the next iteration $C_{n+} = C_n \setminus \{t_a\} \cup \{t_b\}$:

$$error\ ratio = \frac{error(C_{n+})}{error(C_n)} = \frac{\sum_{g \in G} \sum_{r \in R} \sum_{t \in T \setminus C_{n+}} (\hat{e}_{gt}^{C_{n+}} - e_{gt}^r)^2}{\sum_{g \in G} \sum_{r \in R} \sum_{t \in T \setminus C_n} (\hat{e}_{gt}^{C_n} - e_{gt}^r)^2} \tag{1}$$

where $\hat{e}_{gt}^{C_n}$ is our spline based estimate of the expression of gene $g$ at time $t$ by fitting smoothing spline over points $C_n$. If there are pairs which lead to an error ratio of less than 1 in the above function, we select the best (lowest error), assign it to $C_{n+1}$ and continue the iterative process. Otherwise we terminate the process and output $C_n$ as the optimal solution. While the process is guaranteed to converge, given the large combinatorial search space convergence can be slow. This makes adequate initialization an important issue which we have focused on. In practice we find that the search usually converges very fast (within $10 - 15$ iterations).

- *Fitting smoothing spline:* The third key step of our approach is fitting a smoothing spline to every gene independently for the selected subset of time points. As discussed above, this is done by using a regularized version of approximating splines which allow us to determine a unique number of control points and spacing for each of the genes. See Appendix Methods for more details.

## Individual vs. cluster-based evaluation

So far, we assumed that error of each gene has the same contribution to the overall error. However, this assumption ignores the fact that the expression profiles of genes are correlated with the expression of other genes. To take the correlation between gene profiles into account, we also performed cluster based evaluation of genes where we analyzed the error by weighting each gene in terms of inverse of the numbers of genes in the cluster it belongs. This scheme ensures that each cluster contributes equally to the resulting error rather than each gene. We find clusters by k-means algorithm over time series-data by treating each gene as a point in $R^T$ space as well as over a vector of randomly sampled $T$ time points on fitted spline (*Bishop, 2006*). We use Bayesian Information Criterion (BIC) to determine the optimal number of clusters (*Schwarz, 1978*).

# Acknowledgement

We thank the LungMAP consortium for useful comments regarding the methods and analysis presented in this paper. Work supported in part by NIH grant U01 HL122626.

# Additional information

### Funding

| Funder | Grant reference number | Author |
|---|---|---|
| National Institutes of Health | U01 HL122626 | Ziv Bar-Joseph |

The funders had no role in study design, data collection and interpretation, or the decision to submit the work for publication.

### Author contributions

MK, Software, Formal analysis, Validation, Investigation, Visualization, Methodology, Writing—original draft, Writing—review and editing; ES, Conceptualization, Formal analysis, Validation, Investigation, Visualization, Methodology, Writing—original draft; TN, CE, DC, JSH, NK, NA, Resources, Data curation, Investigation; ZB-J, Conceptualization, Resources, Supervision, Funding acquisition, Investigation, Methodology, Writing—original draft, Project administration, Writing—review and editing

### Author ORCIDs

Ziv Bar-Joseph, http://orcid.org/0000-0003-3430-6051

### Ethics

Animal experimentation: This study was performed in strict accordance with the recommendations in the Guide for the Care and Use of Laboratory Animals of the National Institutes of Health. All of the animals were handled according to approved institutional animal care and use committee (IACUC) protocols (APN 10042) of the University of Alabama at Birmingham. All lungs were isolated immediately following euthanasia using approved protocols.

# Additional files

### Supplementary files

• Supplementary file 1. Raw mRNA expression values for the 126 genes studied using nanostring

• Supplementary file 2. Raw miRNA expression values from the nanostring analysis.

### Major datasets

The following dataset was generated:

|  | Database, license, and accessibility |
|---|---|

| Author(s) | Year | Dataset title | Dataset URL | information |
|---|---|---|---|---|
| Kleyman M, Sefer E, Nicola T, Espinoza C, Chhabra D, Hagood JS, Kaminski N, Ambalavanan N, Joseph ZB | 2016 | miRNA Data of Mouse Lung Developement | http://sb.cs.cmu.edu/TPS/data.html | Publicly available at the Systems Biology Group, School of Computer Science, Carnegie Mellon University website |

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

## Appendix 1 Methods

### Selecting the set of 126 genes

*Table 1* provides the list of genes used for the nanostring analysis and the rational for their inclusion.

### DNA Methylation analysis

Mouse alveolar lung tissues attached to LCM caps were stored at −80°C until processing. DNA was extracted using the ZR Genomic DNA-Tissue MicroPrep kit (Zymo Research). Incubation with Digestion buffer and proteinase K was done overnight at 55°C in inverted tubes. 13 genes were chosen for targeted NextGen bisulfite sequencing (NGBS): *Igfbp3, Wif1, Cdh11, Eln, Sox9, Tnc, Dnmt3a, Akt, VEGF, Lox, FoxF2, ZFP536* and *Src*, based on published data (*Cuna et al., 2015*). The presence of CpG islands in 5-UTR, gene body and 3-UTR was interrogated using NCBI Epigenomics database, as well as CpG island searcher (*Takai and Jones, 2002*), and EMBOSS Cpgplot (*Rice et al., 2000*). Targeted NGBS was done by Epigendx Inc. Gene sequences from selected regions were acquired from the Ensembl database. Gene IDs, transcript IDs, simplex PCR IDs, and target regions for each gene are listed in *Appendix 2—Table 3*. A total of 42 target PCRs were designed by PyroMark Assay Design Software (Qiagen).

Targeted NGBS was done on samples collected at the following time points: E16.5, E18.5, P0.5, P1.5, P2.5, P5, P10, P15, P19 and P26. Mouse genomic DNA (200–500 ng) was bisulfite treated using the EZ DNA Methylation Kit (Zymo Research). Multiplex PCR was performed using 0.5 units of TaKaRa EpiTaq HS (Takara Bio) in 2x master mix.

FASTQ files were aligned using open source Bismark Bisulfite Read Mapper using Bowtie2. Methylation levels were calculated in Bismark by dividing the number of methylated reads by the number of total reads, considering all CpG sites covered by a minimum of 30 total reads. Sites where the difference in methylation was less than 5% over the entire time period, those where there was a difference of >20% at a single time point and those with less than 3 non zero values were removed from the analyses.

### TPS Algorithm

A pseudocode for the TPS algorithm is presented in Algorithm 1.3.

---

Algorithm 1. *TPS* : Iterative $k$-point selection

---

1: **Procedure** Iterative–Temporal–Selection
2:　　$C_0 =$ select initial $k$ time points by absolute difference sorting
3:　　$e_0 =$ error of remaining points by fitting splines to $C_0$
4:　　$i = 0$
5:　　**do**
6:　　　　**for** each pair $(t_a, t_b) \in (T^- \setminus C_i) \times C_i$ **do**
7:　　　　　　$C^* = C_i \cup \{t_a\} \{t_b\}$
8:　　　　　　$e^* =$ estimate error by fitting smoothing spline to $C^*$ where regularization parameter is estimated by LOOCV
9:　　　　　　**if** $e^* < e_i$ **then**
10:　　　　　　　$C_{i+1} = C^*$
11:　　　　　　　$e_{i+1} = e^*$
12:　　　　　**end if**
13:　　　　　$i = i + 1$
14:　　　　**end for**
15:　　**While** $e_{i+1} < e_i$
16:　　Output $C_i$ and $e_i$
17: **end procedure**

---

## Selecting the initial set of points

When using an iterative algorithm to solve non-convex problems with several local minima, a key issue is the appropriate selection of the initial solution set. We have tested a number of methods for performing such initializations. The simplest method we tried is to uniformly select a subset of the points (so if $k = T/4$ we use each 4'th point). Another method we tested is to partition the set of all time points $T$ into $k − 1$ intervals of almost equal size. This method determines these boundaries by estimating the cumulative number of points until each time point and selecting time points with cumulative values $\frac{T}{k-1}, 2\frac{T}{k-1}, \ldots, (k − 2)\frac{T}{k-1}$ respectively. Then, it uses $k$ interval boundaries including $t_1$ and $t_T$ as initial solution. We also tested a method that relies on the changes between consecutive time points to select the most important ones for our initial set. Specifically, we sort all points except $t_1$ and $t_T$ by average absolute difference with respect to its predecessor and successor time points by computing:

$$m_{t_i} = \frac{\sum_{g \in G} |Md(e_{gt_{i-1}}) - Md(e_{gt_i})| + |Md(e_{gt_{i+1}}) - Md(e_{gt_i})|}{2|G|} \quad (2)$$

where $Md(e_{gt_i})$ is the median expression for gene $g$ at time $t_i$. We then select the $k − 2$ points with maximum $m_{t_i}$ as the initial solution.

Finally, we developed an alternative initialization method, based on dyanmic recalculation of a metric on each time point. Metric A is same equal to the equation shown above. Metric B of a time point is the difference absolute difference with respect to its predecessor and successor time points. Metric C of a time point is absolute difference with respect to only its predecessor. The alternative initialization algorithm calculates the given metric on each time point other than the first and last and then places those points in a min heap based on the metric. The top(minimum) point in the heap is removed. The metric is recalculated for the point's predeccesor and succesor based on thier neighboring points, using only the points remaining in the heap. This process is repated until only k-2 time points remain in the heap. Then the first time point, last time point and the points remaind in the heap are chosen.

$$MetricA_{e,t_i} = \frac{\sum_{g \in G} |(Md(e_{gprevious_{t_i}}) - Md(e_{gt_i})) + (Md(e_{gnext_{t_i}}) - Md(e_{gt_i}))|}{2|G|} \quad (3)$$

$$MetricB_{e,t_i} = \frac{\sum_{g \in G} |(Md(e_{gprevious_{t_i}}) - Md(e_{gt_i})) - (Md(e_{gnext_{t_i}}) - Md(e_{gt_i}))|}{2|G|} \quad (4)$$

$$MetricC_{e,t_i} = \frac{\sum_{g \in G} |(Md(e_{gprevious_{t_i}}) - Md(e_{gt_i}))}{2|G|} \quad (5)$$

Algorithm 2: Init TPS: Iterative initial $k$ point selection

*continued*

**Algorithm 2: Init TPS: Iterative initial $k$ point selection**

1: **Procedure** Iterative–Initial Point–Selection
2:   $H$ = Empty min heap
3:   $e$ = matrix where rows are genes and columns are time points, values are expression measurements
4:   **for** each time point $t$ (other than the first and last) **do**
5:     $value_t = Metric_{e,t}$
6:     $previous_t = t - 1$
7:     $next_t = t + 1$
8:     Add $value_t$ to $H$
9:   **end for**
10:  **While** $size(H) > k - 2$ **do**
11:     Remove minimum $value_m$ time point $m$ from $H$
12:     $previous_{next_m} = previous_m$
13:     $next_{previous_m} = next_m$
14:     Remove $value_{previous_m}$ from $H$
15:     Remove $value_{next_m}$ from $H$
16:     Remove $m$ from $e$
17:     $value_{previous_m} = Metric_{e,previous_m}$
18:     $value_{next_m} = Metric_{e,next_m}$
19:     Add $value_{next_m} to H$
20:     Add $value_{previous_m} to H$
21:  **end while**
22:  Ouput all $t$ left as $value_t$ in $H$ + first time point + last time point
23: **end procedure**

We found that for our particular dataset, the dynamic initialization with $MetricA_{e,t_i}$ performed best for selections of time points smaller than one third of the the initial dense time series, while the non dynamic $m_{t_i}$ method works best for selections of time points between one third and and one half of the initial time series. The dyanmic metric and non dynamic metrics can be compared in their performance on our data in *Figure 1—figure supplement 2*. However, all of the metrics performed much better than a selection of random points as shown in *Figure 1—figure supplement 3*.

# Further improvements to the iterative points selection procedures

We tested the following possible search strategies to improve the iterative points removal and addition in TPS.

- We add and remove $b$ time points in each iteration instead of a single point. This increases the complexity of each iteration from $O(kGT^2Q)$ to $O(kGT^{2b}Q)$ where $Q$ is the complexity of fitting a smoothing spline.
- We use simulated annealing to escape from local minima (*Kirkpatrick et al., 1983*). In this case, we do not always move to a pair of points with the minimum error in each iteration, but instead move to a solution with random pair of points with probability 1 if its error $e^r$ is lower than error of current solution $e^i$ whereas we move to a solution with probability $e^{-C(e^r - e^i)}$ if $e^r \geq e^i$. Here, $C$ is the temperature that increases by increasing number of iterations and the probability of moving to a solution with larger error decreases over time.

In practice, even though both approaches should in theory be better able to escape local minima than the greedy approach described above, for the data we analyzed they do not perform significantly better as *Figure 2* in the main text demonstrates.

## Fitting smoothing spline

*TPS* uses splines for fitting expression curves. Regularized smoothing spline satisfies the piecewise cubic polynomial $\mu(t) = a_i + b_i(t - t_i) + c_i(t - t_i)^2 + d_i(t - t_i)^3$ for $t \in [t_i, t_{i+1})$, $i \in 1, \ldots, T - 1$ as shown in **Wahba (1990)**. Then, according to (**Reinsch, 1967**; **De Boor, 1978**), regularized smoothing spline objective can also be expressed as in:

$$\min (y - a)^{'}(y - a) + \lambda c^{'} R c \tag{6}$$

where $a = (a_1, a_2, \ldots, a_T)$, $c = (c_2, c_3, \ldots, c_{T-1})$, and $R$ is a $(n - 2)^2$ tridiagonal symmetric matrix with entries $r_{i,i} = \frac{2(h_i + h_{i+1})}{3}$, $r_{i,i+1} = \frac{h_{i+1}}{3}$ where $h_i = t_{i+1} - t_i$. The continuity restrictions imply that:

$$R c = Q^{'} a \tag{7}$$

where $Q$ is an $n \times (n - 2)$ tridiagonal matrix with entries $q_{i,i+1} = \frac{1}{h_{i+1}}$, $q_{i+1,i} = \frac{1}{h_{i+1}}$ and $q_{i,i} = -(\frac{1}{h_i} + \frac{1}{h_{i+1}})$. Thus, we may write **Equation 6** as:

$$\min (y - a)^{'}(y - a) + \lambda a^{'} Q R^{-1} Q^{'} a \tag{8}$$

where $a$ can be derived as in:

$$a = (I + \lambda Q R^{-1} Q^{'})^{-1} y \tag{9}$$

Once $a$ is estimated, $b$, $c$, $d$ are estimated by corresponding Equations in **Reinsch (1967)**.

For our specific setting, we also introduce a regularization parameter to enable us to determine the number of control points. Let $I_g = \{(t, Md(e_{gt})), t \in C\}$, and $\mu$ be the spline we are interested in fitting, smoothing spline can be found by the following optimization problem which minimizes penalized least-squares error:

$$\min \sum_{(t, y_t) \in I_g} (y_t - \mu(t))^2 + \lambda \int_{t_1}^{t_T} \mu^{''}(x)^2 dx \tag{10}$$

where $\lambda$ is the regularization parameter which prevents overfitting by affecting the number of knots selected. We estimated $\lambda$ by leave-one-out cross-validation (LOOCV) in our experiments (See Appendix Methods for details of smoothing spline fitting).

## Proteomics analysis

Proteins were extracted using tissue protein extraction reagent (T-PER, Thermo) as per manufacturer's instructions, carried out directly on the micro-dissection cap. Protein concentrations was determined with the EZQ protein assay (Life Sciences). The proteins were digested overnight at 37C, followed by acidification to pH $3 - 4$ with $10\%$ formic acid (FA), and extracted as per manufacturer's instructions, then concentrated to near completion using a Savant SpeedVac Concentrator (Thermo) and diluted with $0.1\%$ FA to a final concentration of $\sim 100$ ng \uL for analysis by LCMS. The LCMS data were converted to a universal MzXML file format prior to being searched using SEQEST (Thermo) against a Mouse subset of the UniRef100 database. These data were then uploaded to Scaffold (Proteome Software) in order to filter and group each peptide ID to specific proteins with peptide probability scores set at $80\%$, and protein probability scores set at $99\%$. Using only proteins presenting with 2 or more peptides per protein, the confidence interval was set to

~99.9% with and FDR <0.1. Quantification was carried out using Scaffold Q + using normalized spectral counts.

## Appendix 2 Results

### Example of a *TPS* run

Here we discuss a specific setting for *TPS* that allows us to discuss the set of points selected and their relevance. Specifically, to test *TPS* , we fixed three set points in advance (first (0.5'th day) and last (28'th day), which are required for any setting and day 7 which was previously determined to be of importance to lung development. Next, we have asked TPS to further select 10 more points (for a total of 13). For this setting, the method selected the following points: 0.5, 1.0, 1.5, 2.5, 4, 5, 7, 10, 13.5, 15, 19, 23, 28. While we do not know the ground truth, the larger focus on the earlier time points determined by the method (with 7 of the 13 points for the first 7 days) makes sense in this context as several aspects of lung differentiation are determined in the first week (*Guilliams et al., 2013*). The other 3 weeks were more or less uniformly sampled by our *TPS* . This highlights the usefulness of an unbiased approach to sampling time points rather than just uniformly sampling through the time window.

### *TPS* identifies subset of important time points across multiple genes

To understand whether gene-expression profiles over time has a simple trend, we also compare the reconstruction performance of *TPS* with fitting piecewise linear curves between initial and middle time points and between middle and last time points. The reconstruction error by *TPS* is significantly better than the piecewise linear reconstruction for 102 genes out of 126 genes. We have plotted the comparison of reconstruction for several of these genes as in *Figure 2— figure supplement 3*. The distribution of error difference between these methods looks significantly different than normal distribution ($p<0.0001$ by Shapiro-Wilk test).

### miRNA clusters are enriched for several biological processes

While the mRNA datasets includes only a handful of genes (less than 0.5% of all genes) the miRNA data includes more profiles and so further analysis of this data can be perfromed. We have performed clustering of the miRNA data using k-means (*Hartigan, 1975*) where the number of clusters is selected by Bayesian Information Criteria (*Schwarz, 1978*) leading to 8 stable miRNA clusters *Figure 4—figure supplement 2*. Next, we mapped miRNA's to predicted targets using TargetScan (*Agarwal et al., 2015*), and performed gene-enrichment analysis by FuncAssociate (*Berriz et al., 2003*). We find clusters to be enriched for several Gene Ontology biological processes (*Ashburner et al., 2000*). For instance, cluster 4 is enriched for single-organism cellular process, positive regulation of biological process, regulation of metabolic process, etc. See Supporting Website for complete results.

### miRNA reconstruction

*Figure 4—figure supplement 1* presents the reconstructed and measured expression values for a few miRNAs based on time points identified using the mRNA dataset. Several of these miRNAs are known to be involved in regulation of lung development. For example, *mmu-miR-100* is known to regulate *Fgfr3* and *Igf1r*, *mmu-miR-136* targets *Tgfb2*, *mmu-miR-152* targets *Meox2*, *Robo1*, *Fbn1*, *Nfya* (*Popova et al., 2014*). Additional figures for all miRNAs and mRNAs are avialable on the supporting website.

## TPS application to select time points for proteomics analysis

We used mass spectrometry to profile the levels of 1020 proteins over the optimal 13 time points determined by *TPS* (using the mRNA expression data):
[0.5, 1.0, 1.5, 2.5, 4.0, 5.0, 7.0, 10.0, 13.5, 15.0, 19.0, 23.0, 28.0]). To test the ability of *TPS* to determine the optimal time points for the proeomics data (based only on the mRNA data) we performed a similar analysis to the analysis performed for the miRNA data. Specifically, we used *TPS* to select subset of 4 to 12 of these points *based on the mRNA data* and compared the error using these points to random and uniform selection of the same number of points. The results are presented in . In addition to comparing *TPS* to random and uniform we have also compared different strategies for initializing the set of points as discussed in Method. Finally, the figure also presents the repeat noise values which is the theoretical limit for the performance of any profile reconstruction method.

As for the miRNA data, we see a significant and consistent improvement (for all number of selected time points) over uniform sampling highlighting the advantage of condition specific sampling decisions. Again, as the number of points used by *TPS* increases, it leads to results that are very close to the error represented by noise in the data (17.47).

## Analysis of methylation data

Methylation data included 3 repeats for time points 0.5, 1.5, 2.5, 5, 10, 15, 19, 26 for 266 loci belonging to 13 genes. Among these genes all except *Zfp536* were also profiled in our nanostring mRNA analysis. *Appendix 2—Table 2* summarizes the number of loci for each gene in the methylation dataset. We used shifted percentage of methylation at each time point in our analysis which is obtained by subtracting the median percentage of methylation at initial time point (baseline) from all data points for each gene. *Figure 5—figure supplement 2* presents the best positive or negative correaltion observed between the methylation data and the gene expression data for these genes (note that we do not expect all up stream regions to show a correlated profile since it is likely that only a subset, or even a single, region is responsible for the changes in expression observed which is why we look for the most correlated or anti-correlated region).

## Importance of correct determination of expression profiles

As shown in *Figure 6* in the main text, TPS results differ from prior methods when reconstructing expression profiles for several genes. Below we discuss the significance of these differences and their impact on the ability to correctly assign function to that gene:

- *Nol3*: Nucleolar protein 3 (apotosis repressor with CAR domain) gene (also called *ARC*) encodes a protein that inhibits apoptosis, by decreasing activities of Caspases 2 and 8 and tumor protein p53. Evaluation of the TPS profile suggests that the increase in Nol3 correlates with postnatal lung development, with a rapid increase from birth until 2 weeks of age, followed by stabilization, while the prior sampling rates show only an initial peak and then decrease. While the exact role of *Nol3* in lung development has not been established, it is known that *Nol3* protects pulmonary arterial smooth muscle cells from hypoxia-induced death and facilitates growth factor-induced proliferation and hypertrophy, and is probably involved in human pulmonary hypertension (*Turi et al., 1990*). *Nol3* is a regulator of myogenic differentiation (*Hunter et al., 2007*) and its pattern of expression suggests that it may be important in regulating pulmonary airway and vascular smooth muscle development and differentiation.

- *Esr2*: The gene estrogen receptor beta encodes a receptor for estrogen, and is important in regulating lung development and modulating differences in lung development between males and females (**Gortner et al., 2013**). Evaluation of the TPS profile suggests that the *Esr2* decreases briefly after birth, followed by an increase from around day 5 until day 20 whereas non-optimized profile suggests a relatively flat profile. While fetal mouse lungs express both *Esr2* alpha and beta, adult mouse lungs express only *Esr2* beta consistent with the TPSresults (**Carvalho and Goncalves, 2012**).

- *Igfbp3*: Insulin-like growth factor binding protein 3 ( *Igfbp3*) belongs to the Igfbp family and has a Igfbp domain and a thyroglobulin type-I domain (http://www.ncbi.nlm.nih.gov/gene/3486). The TPS profile for *Igfbp3* is very different from the non-optimized profile, suggesting that important biological information is lost when not using the TPS profile. *Igfbp3* regulates the induction of TNC by TGF-beta (**Brissett et al., 2012**) and both these molecules are critical in lung alveolar septation.

- *Wif1*: *Wnt* inhibitory factor 1 ( *WIF1*) inhibits Wnt proteins, that are well known to be critical in many stages of lung development. The TPS profile is very different from the non-optimized profile, as the TPS profile indicates a much earlier and higher peak of *WIF1* during postnatal lung development that may be critical in alveolar septation. *WIF1* is a target gene for Smad1, one of the *BMP* receptor proteins important in lung development and maturation. A regulatory loop of *Bmp4-Smad1-Wif1-Wnt/beta-catenin* may coordinate *BMP* and *Wnt* pathways to control lung development (**Xu et al., 2011**), and dysregulation of the *Smad1/Wif1* axis is associated with lung hypoplasia (**Fujiwara et al., 2012**).

- *Inmt*: Indolethylamine N-methyl transferase (*Inmt*) gene encodes an enzyme that N-methylates indoles such as tryptamine (http://www.ncbi.nlm.nih.gov/gene/11185). The TPS profile for *Inmt* is very different from the non-optimized profile, as the TPS profile indicates a much lower and prolonged reduction of *Inmt* during postnatal lung development. Methyl conjugation is an important pathway in the metabolism of many drugs, neurotransmitters, and xenobiotic compounds (**Thompson and Weinshilboum, 1998**). While it is known that *Inmt* expression varies over the course of human lung development (**Kopantzev et al., 2008**), its exact role in lung development is not known.

- *Fgf18*: Fibroblast growth factor 18 (*Fgf18*) is a member of the fibroblast growth factor family, and the Fgfs are well known to be critical in multiple stages of lung development. The non-optimized profile indicates a smaller and later peak, and is not similar to the TPS profile which suggests a much more improtant role.*Fgf18* is a pleiotropic growth factor that stimulates proliferation in a number of tissues (http://www.ncbi.nlm.nih.gov/gene/8817). *Fgf18* is highly expressed in the developing lung as the TPS profile indicates (**Ohbayashi et al., 1998**), and Fgfr3 is important in postnatal alveolar development (**Weinstein et al., 1998**). The role of *Fgf18* in regulating fibroblast proliferation (**Hu et al., 1999**) may be important in alveolar septation, as *Fgf18* increases after birth with a peak around P10, with reduction after completion of alveolar septation.

**Appendix 2—table 1.** List of genes used for the Nanostring analysis and the rational for their inclusion.

| Ensembl gene ID | Accession number | Gene name | Rationale |
| --- | --- | --- | --- |
| ENSMUSG00000024130 | NM_001039581.2 | Abca3 | Alveolar Type II cell marker |
| ENSMUSG00000031378 | NM_007435.1 | Abcd1 | important in other processes (IPF, COPD etc) |
| ENSMUSG00000029802 | NM_011920.3 | Abcg2 | Mesenchymal cell marker |
| ENSMUSG00000035783 | NM_007392.3 | Acta2 | Fibroblast cell marker |
| ENSMUSG00000029580 | NM_007393.1 | Actb | Common house-keeping gene |
| ENSMUSG00000036040 | NM_029981.1 | Adamtsl2 | Altered DNA methylation during septation |
| ENSMUSG00000015452 | NM_007425.2 | Ager | Alveolar Type I cell marker |
| ENSMUSG00000001729 | NM_001165894.1 | Akt1 | Altered DNA methylation during septation |
| ENSMUSG00000053279 | NM_013467.3 | Aldh1a1 | Important for septation |
| ENSMUSG00000013584 | NM_009022.3 | Aldh1a2 | Potentially important for septation |
| ENSMUSG00000022244 | NM_008537.4 | Amacr | important in other processes (IPF , COPD etc) |
| ENSMUSG00000044217 | NM_009701.4 | Aqp5 | Alveolar Type I cell marker |

*Appendix 2—table 1 continued on next page*

*Appendix 2—table 1 continued*

| Ensembl gene ID | Accession number | Gene name | Rationale |
| --- | --- | --- | --- |
| ENSMUSG00000026576 | NM_009721.5 | Atp1b1 | Lung fluid clearance |
| ENSMUSG00000060802 | NM_009735.3 | B2m | Common house-keeping gene |
| ENSMUSG00000102037 | NM_009742.3 | Bcl2a1a | Apoptosis regulator |
| ENSMUSG00000056216 | NM_009884.3 | Cebpg | Important for lung development |
| ENSMUSG00000029084 | NM_007646.4 | Cd38 | Airway smooth muscle cell functional responses |
| ENSMUSG00000018774 | NM_009853.1 | Cd68 | Monocyte cell marker |
| ENSMUSG00000031673 | NM_009866.4 | Cdh1 | Epithelial cell marker |
| ENSMUSG00000064246 | NM_007695.2 | Chil1 | Monocyte cell marker |
| ENSMUSG00000040809 | NM_009892.1 | Chil3 | Increased during septation |
| ENSMUSG00000022512 | NM_016674.3 | Cldn1 | Tight junction protein |
| ENSMUSG00000070473 | NM_009902.4 | Cldn3 | Tight junction protein (mostly epithelial) |
| ENSMUSG00000041378 | NM_013805.4 | Cldn5 | Tight junction protein |
| ENSMUSG00000018569 | NM_016887.6 | Cldn7 | Tight junction protein (mostly epithelial) |
| ENSMUSG00000001506 | NM_007742.3 | Col1a1 | Fibroblast cell marker |
| ENSMUSG00000063063 | NM_009819.2 | Ctnna2 | Altered DNA methylation during septation |
| ENSMUSG00000031360 | NM_001168571.1 | Ctps2 | important in other processes (IPF , COPD etc) |
| ENSMUSG00000040856 | NM_010052.4 | Dlk1 | Decreased during septation |
| ENSMUSG00000020661 | NM_007872.4 | Dnmt3a | Altered DNA methylation during septation |
| ENSMUSG00000046179 | NM_001013368.5 | E2f8 | Altered DNA methylation during septation |
| ENSMUSG00000000303 | NM_009864.2 | Cdh1 | Epithelial cell marker |
| ENSMUSG00000020122 | NM_207655.2 | Egfr | Important for lung development |
| ENSMUSG00000029675 | NM_007925.3 | Eln | Altered DNA methylation during septation |
| ENSMUSG00000045394 | NM_008532.2 | Epcam | Epithelial cell marker |
| ENSMUSG00000052504 | NM_010140.3 | Epha3 | Involved in lung development |
| ENSMUSG00000028289 | NM_001122889.1 | Epha7 | Involved in lung cancer, potential role in development |
| ENSMUSG00000021055 | NM_010157.3 | Esr2 | Important regulator of multiple processes |
| ENSMUSG00000061731 | NM_010162.2 | Ext1 | Altered DNA methylation during septation |
| ENSMUSG00000039109 | NM_001166391.1 | F13a1 | Involved in lung injury , cancer |
| ENSMUSG00000057967 | NM_008005.1 | Fgf18 | Important for septation |
| ENSMUSG00000030849 | NM_010207.2 | Fgfr2 | Important regulator of multiple processes |
| ENSMUSG00000078302 | NM_008242.2 | Foxd1 | Pericyte cell marker |
| ENSMUSG00000042812 | NM_010426.1 | Foxf1 | Involved in lung development |
| ENSMUSG00000038402 | NM_010225.1 | Foxf2 | Altered DNA methylation during fibrosis |
| ENSMUSG00000001020 | NM_011311.1 | S100a4 | Fibroblast cell marker |
| ENSMUSG00000057666 | NM_001001303.1 | Gapdh | Common house-keeping gene |
| ENSMUSG00000005836 | NM_010258.3 | Gata6 | Important regulator of multiple processes |
| ENSMUSG00000029992 | NM_013528.3 | Gfpt1 | important in other processes (IPF, COPD etc) |
| ENSMUSG00000041624 | NM_001033322.2 | Gucy1a2 | Important for septation |
| ENSMUSG00000025534 | NM_010368.1 | Gusb | Common house-keeping gene |
| ENSMUSG00000021109 | NM_010431.2 | Hif1a | Hypoxia signaling |
| ENSMUSG00000058773 | NM_020034.1 | Hist1h1b | Decreased during septation |
| ENSMUSG00000061615 | NM_175660.3 | Hist1h2ab | Decreased during septation |

*Appendix 2—table 1 continued*

| Ensembl gene ID | Accession number | Gene name | Rationale |
| --- | --- | --- | --- |
| ENSMUSG00000032126 | NM_013551.2 | Hmbs | Common house-keeping gene |
| ENSMUSG00000029919 | NM_019455.4 | Hpgds | important in other processes (IPF, COPD etc) |
| ENSMUSG00000025630 | NM_013556.2 | Hprt | Common house-keeping gene |
| ENSMUSG00000020053 | NM_001111274.1 | Igf1 | Regulating miRNA altered during septation |
| ENSMUSG00000020427 | NM_008343.2 | Igfbp3 | Altered DNA methylation during septation, fibrosis |
| ENSMUSG00000003477 | NM_009349.3 | Inmt | Increased during septation |
| ENSMUSG00000026768 | NM_001001309.2 | Itga8 | Involved in lung development |
| ENSMUSG00000040029 | NM_001081113.1 | Ipo8 | important in other processes (IPF, COPD etc) |
| ENSMUSG00000030786 | NM_001082960.1 | Itgam | Monocyte cell marker |
| ENSMUSG00000030789 | NM_021334.2 | Itgax | Monocyte cell marker |
| ENSMUSG00000090122 | NM_021487.1 | Kcne1l | important in other processes (IPF, COPD etc) |
| ENSMUSG00000063142.10 | XM_006518608.1 | Kcnma1 | Altered DNA methylation during septation |
| ENSMUSG00000079852 | NM_010649.3 | Klra4 | Increased during septation |
| ENSMUSG00000023043 | NM_010664.2 | Krt18 | Epithelial cell marker |
| ENSMUSG00000061527 | NM_027011.2 | Krt5 | Basal cell marker |
| ENSMUSG00000029570 | NM_008494.3 | Lfng | Important for septation |
| ENSMUSG00000024529 | NM_010728.2 | Lox | Altered DNA methylation during fibrosis |
| ENSMUSG00000028003 | NM_023624.4 | Lrat | Increased during septation |
| ENSMUSG00000027070 | NM_001081088.1 | Lrp2 | Altered DNA methylation during septation |
| ENSMUSG00000061068 | NM_010779.2 | Mcpt4 | Decreased during septation |
| ENSMUSG00000026110 | NM_173870.2 | Mgat4a | Involved in acute lung injury |
| ENSMUSG00000043613 | NM_010809.1 | Mmp3 | Increased during septation |
| ENSMUSG00000018623 | NM_010810.4 | Mmp7 | Important in lung fibrosis |
| ENSMUSG00000066108 | XM_006508653.1 | Muc5b | Important in lung fibrosis |
| ENSMUSG00000037974 | NM_010844.1 | Muc5ac | Epithelial cell marker |
| ENSMUSG00000024304 | NM_007664.4 | Cdh2 | Tight Junction/Adhesion |
| ENSMUSG00000054008 | NM_008306.4 | Ndst1 | Involved in pathologic airway remodeling |
| ENSMUSG00000031902 | NM_010901.2 | Nfatc3 | Important for lung development |
| ENSMUSG00000073435 | NM_019730.2 | Nme3 | Apoptosis-related gene |
| ENSMUSG00000026575 | NM_138314.3 | Nme7 | Important for stem cell renewal |
| ENSMUSG00000014776 | NM_030152.4 | Nol3 | Regulating miRNA altered during septation |
| ENSMUSG00000051048 | NM_177161.4 | P4ha3 | Important in lung fibrosis |
| ENSMUSG00000068039 | NM_013686.3 | Tcp1 | Basal cell marker |
| ENSMUSG00000029998 | NM_025823.4 | Pcyox1 | important in other processes (IPF , COPD etc) |
| ENSMUSG00000029231 | NM_011058.2 | Pdgfra | Important for septation |
| ENSMUSG00000024620 | NM_008809.1 | Pdgfrb | Pericyte cell marker |
| ENSMUSG00000028583 | NM_010329.2 | Pdpn | Alveolar Type I cell marker |
| ENSMUSG00000062070 | NM_008828.2 | Pgk1 | important in other processes (IPF , COPD etc) |
| ENSMUSG00000053398 | NM_016966.3 | Phgdh | important in other processes (IPF, COPD etc) |
| ENSMUSG00000005198 | NM_009089.2 | Polr2a | important in other processes (IPF, COPD etc) |
| ENSMUSG00000071866 | NM_008907.1 | Ppia | Common house-keeping gene |
| ENSMUSG00000024997 | NM_007452.2 | Prdx3 | Mitochondrial oxidative stress regulator |

*Appendix 2—table 1 continued on next page*

*Appendix 2—table 1 continued*

| Ensembl gene ID | Accession number | Gene name | Rationale |
|---|---|---|---|
| ENSMUSG00000026134 | NM_008922.2 | *Prim2* | Expressed in placenta and crucial for mammalian growth. |
| ENSMUSG00000033491 | NM_178738.3 | *Prss35* | Decreased during septation |
| ENSMUSG00000032487 | NM_011198.3 | *Ptgs2* | Regulating miRNA altered during septation |
| ENSMUSG00000056458 | NM_011973.2 | *Mok* | Alveolar Type I cell marker |
| ENSMUSG00000037992 | NM_001177302.1 | *Rara* | Important for septation |
| ENSMUSG00000022883 | NM_019413.2 | *Robo1* | Altered DNA methylation during septation |
| ENSMUSG00000025508 | NM_026020.6 | *Rplp2* | |
| ENSMUSG00000066361 | NM_008458.2 | *Serpina3c* | Increased during septation |
| ENSMUSG00000022097 | NM_011359.1 | *Sftpc* | Alveolar Type II cell marker |
| ENSMUSG00000021795 | NM_009160.2 | *Sftpd* | Alveolar Type II cell marker |
| ENSMUSG00000050010 | NM_001033415.3 | *Shisa3* | Altered DNA methylation during septation |
| ENSMUSG00000032402 | NM_016769.3 | *Smad3* | Important for septation |
| ENSMUSG00000042821 | NM_011427.2 | *Snai1* | Important for lung development and injury |
| ENSMUSG00000000567 | NM_011448.4 | *Sox9* | Altered DNA methylation during septation |
| ENSMUSG00000027646 | NM_001025395.2 | *Src* | Altered DNA methylation during septation |
| ENSMUSG00000014767 | NM_013684.3 | *Tbp* | Common house-keeping gene , involved in multiple processes |
| ENSMUSG00000000094 | NM_172798.1 | *Tbx4* | Altered DNA methylation during septation |
| ENSMUSG00000032228 | NM_011544.3 | *Tcf12* | Involved in multiple developmental processes |
| ENSMUSG00000022797 | NM_011638.3 | *Tfrc* | Common house-keeping gene |
| ENSMUSG00000002603 | NM_011577.1 | *Tgfb1* | Important for septation |
| ENSMUSG00000045691 | NM_153083.5 | *Thtpa* | important in other processes (IPF, COPD etc) |
| ENSMUSG00000032011 | NM_009382.3 | *Thy1* | Fibroblast cell marker |
| ENSMUSG00000028364 | NM_011607.1 | *Tnc* | Altered DNA methylation during septation |
| ENSMUSG00000044986 | NM_009437.4 | *Tst* | important in other processes (IPF, COPD etc) |
| ENSMUSG00000026803 | NM_009442.2 | *Ttf1* | Important for lung development |
| ENSMUSG00000008348 | NM_019639.4 | *Ubc* | Common house-keeping gene |
| ENSMUSG00000023951 | NM_001025250.3 | *Vegfa* | Angiogenesis; Altered DNA methylation during septation |
| ENSMUSG00000026728 | NM_011701.4 | *Vim* | Mesenchymal cell marker |
| ENSMUSG00000020218 | NM_011915.1 | *Wif1* | Altered DNA methylation during septation |
| ENSMUSG00000022285 | NM_011740.2 | *Ywhaz* | Common house-keeping gene |

**Appendix 2—table 2.** Summary of methylation dataset

| Gene | Number of loci | Gene | Number of loci |
|---|---|---|---|
| *Cdh11* | 14 | *Zfp536* | 16 |
| *Src* | 11 | *Igfbp3* | 34 |
| *Sox9* | 16 | *Wif1* | 21 |
| *Dnmt3a* | 41 | *Vegfa* | 20 |
| *Eln* | 20 | *Tnc* | 4 |
| *Foxf2* | 41 | *Lox* | 17 |
| *Akt1* | 11 | | |

**Appendix 2—table 3.** Target regions for each gene for methylation analysis

| Gene | Ensembl gene ID | Ensembl transcript ID | Assay ID | Target location | Fwd Tm | Rev Tm | % GC | Coordinates (GRCm38/mm10) |
|---|---|---|---|---|---|---|---|---|
| Akt1 | ENSMUSG00000001729 | ENSMUST00000001780 | ADS3333 | 3' UTR | 68 | 65.5 | 31.5 | chr12:112654548–112654709 |
| Akt1 | ENSMUSG00000001729 | ENSMUST00000001780 | ADS3332 | Intron 9/Exon 10 | 68.3 | 69.8 | 38.3 | chr12:112657120–112657273 |
| Cdh11 | ENSMUSG00000031673 | ENSMUST00000075190 | ADS3308 | Intron 3 | 66.8 | 68.3 | 36.9 | chr8:102677609–102677766 |
| Cdh11 | ENSMUSG00000031673 | ENSMUST00000075190 | ADS3318 | Intron 1 | 64.1 | 69.7 | 37 | chr8:102784569–102784722 |
| Cdh11 | ENSMUSG00000031673 | ENSMUST00000075190 | ADS3307 | Promoter | 69.1 | 71.3 | 29.9 | chr8:102785456–102785649 |
| Dnmt3a | ENSMUSG00000020661 | ENSMUST00000020991 | ADS3326 | Promoter | 68.6 | 67.7 | 47.1 | chr12:3806505–3806659 |
| Dnmt3a | ENSMUSG00000020661 | ENSMUST00000020991 | ADS632 | Intron 1 | 64 | 64.7 | 32.2 | chr12:3834382–3834592 |
| Dnmt3a | ENSMUSG00000020661 | ENSMUST00000020991 | ADS3328 | Exon 6/Intron 6 | 64.7 | 64 | 31.8 | chr12:3901545–3901764 |
| Dnmt3a | ENSMUSG00000020661 | ENSMUST00000020991 | ADS3329 | Intron 6 | 66.8 | 66.1 | 25.4 | chr12:3907514–3907765 |
| Eln | ENSMUSG00000029675 | ENSMUST00000015138 | ADS3319 | Intron 16 | 67.1 | 67.9 | 47.8 | chr5:134721191–134721447 |
| Eln | ENSMUSG00000029675 | ENSMUST00000015138 | ADS3309 | Intron 7/Exon 8/Intron 8 | 64.1 | 67.4 | 37.8 | chr5:134729221–134729526 |
| Eln | ENSMUSG00000029675 | ENSMUST00000015138 | ADS024 | Promoter | 65 | 67.4 | 42.6 | chr5:134747412–134747606 |
| Foxf2 | ENSMUSG00000038402 | ENSMUST00000042054 | ADS4505 | Promoter | 63.1 | 65 | 42.5 | chr13: 31625470–31625556 |
| Foxf2 | ENSMUSG00000038402 | ENSMUST00000042054 | ADS4506 | 5-UTR | 65.1 | 64.8 | 37.9 | chr13:31625904–31626093 |
| Foxf2 | ENSMUSG00000038402 | ENSMUST00000042054 | ADS4507 | 3-Downstream | 68 | 68.9 | 28.1 | chr13:31632481–31632716 |
| Igfbp3 | ENSMUSG00000020427 | ENSMUST00000020702 | ADS5134 | 3-Downstream | 69.3 | 69.6 | 32.1 | chr11:7203969–7204208 |
| Igfbp3 | ENSMUSG00000020427 | ENSMUST00000020702 | ADS3301 | Exon 4/Intron 4 | 70.5 | 70 | 33 | chr11:7208306–7208481 |
| Igfbp3 | ENSMUSG00000020427 | ENSMUST00000020702 | ADS5133 | Intron 1 | 68.3 | 68.5 | 26.1 | chr11:7212803–7213043 |
| Igfbp3 | ENSMUSG00000020427 | ENSMUST00000020702 | ADS5132 | Promoter | 67.8 | 69.2 | 28.7 | chr11:7214210–7214499 |
| Lox | ENSMUSG00000024529 | ENSMUST00000171470 | ADS4512 | Exon 2 | 69 | 70.9 | 31.3 | chr18: 52529184–52529315 |
| Lox | ENSMUSG00000024529 | ENSMUST00000171470 | ADS4513 | Exon 4 | 65.7 | 64.8 | 28.5 | chr18:52526887–52527023 |
| Lox | ENSMUSG00000024529 | ENSMUST00000171470 | ADS4511 | Promoter | 64.7 | 66.4 | 21.2 | chr18:52530080–52530216 |
| Sox9 | ENSMUSG00000000567 | ENSMUST00000000579 | ADS796 | Promoter | 61 | 66.1 | 31.4 | chr11:112781641–112781811 |
| Sox9 | ENSMUSG00000000567 | ENSMUST00000000579 | ADS3311 | Intron 1 | 69.7 | 68.5 | 34.7 | chr11:112783358–112783605 |

*Appendix 2—table 3 continued on next page*

Appendix 2—table 3 continued

| Gene | Ensembl gene ID | Ensembl transcript ID | Assay ID | Target location | Fwd Tm | Rev Tm | % GC | Coordinates (GRCm38/mm10) |
|---|---|---|---|---|---|---|---|---|
| Sox9 | ENSMUSG00000000567 | ENSMUST00000000579 | ADS3310 | Exon 3 | 66.4 | 63.1 | 26.2 | chr11:112784760–112784885 |
| Src | ENSMUSG00000027646 | ENSMUST00000109533 | ADS4514 | Intron 1 | 64.8 | 65.9 | 35.9 | chr2:157423925–157424027 |
| Src | ENSMUSG00000027646 | ENSMUST00000109533 | ADS4515 | Intron 4 | 66.5 | 68.8 | 37.6 | chr2:157457351–157457520 |
| Src | ENSMUSG00000027646 | ENSMUST00000109533 | ADS4516 | Exon 14 | 65.5 | 65.6 | 33.7 | chr2:157469741–157469912 |
| Tnc | ENSMUSG00000028364 | ENSMUST00000107377 | ADS3324 | Intron 14 | 63.3 | 62.2 | 23 | chr4:63982645–63982818 |
| Tnc | ENSMUSG00000028364 | ENSMUST00000107377 | ADS3325 | Intron 14 | 62.5 | 61.6 | 20.2 | chr4:63982799–63982986 |
| Tnc | ENSMUSG00000028364 | ENSMUST00000107377 | ADS3323 | Exon 3 | 65 | 67.5 | 35.2 | chr4:64017478–64017721 |
| Tnc | ENSMUSG00000028364 | ENSMUST00000107377 | ADS3322 | Promoter | 62.7 | 63.9 | 26.7 | chr4:64047034–64047149 |
| Vegfa | ENSMUSG00000023951 | ENSMUST00000071648 | ADS3336 | 3-UTR | 67.2 | 66.9 | 32.6 | chr17:46018598–46018735 |
| Vegfa | ENSMUSG00000023951 | ENSMUST00000071648 | ADS3335 | Intron 2/Exon 3 | 64.6 | 63.8 | 29.5 | chr17:46025336–46025620 |
| Wif1 | ENSMUSG00000020218 | ENSMUST00000020439 | ADS3302 | Promoter | 69.4 | 68.7 | 31.3 | chr10:121033395–121033691 |
| Wif1 | ENSMUSG00000020218 | ENSMUST00000020439 | ADS3303 | Intron 4/Exon 5/Intron 5 | 60.9 | 60.1 | 31.8 | chr10:121083800–121083997 |
| Wif1 | ENSMUSG00000020218 | ENSMUST00000020439 | ADS3304 | Exon 10/3-UTR | 66.6 | 67.5 | 24.8 | chr10:121099752–121099973 |
| Zfp536 | ENSMUSG00000043456 | ENSMUST00000056338 | ADS4510 | 3-Downstream | 65.4 | 67.4 | 35.9 | chr7:37473451–37473606 |
| Zfp536 | ENSMUSG00000043456 | ENSMUST00000056338 | ADS4509 | Exon 4 | 68.4 | 69.9 | 35.4 | chr7:37567973–37568130 |

**Appendix 2—table 4.** Mean and standard deviation of mean squared error over all 126 genes by TPS selecting 5 points and piecewise linear fits over 3 sets of points identified heuristically in the literature.

| Method | Mean | Std dev |
|---|---|---|
| TPS (0.5, 6, 9.5, 19 and 28) | 0.40306335962 | 0.2206665163 |
| Piecewise linear over 0.5, 7, 14, 28 | 0.594072719494 | 0.399642079492 |
| Piecewise linear over 0.5, 2, 14, 28 | 0.710967061349 | 0.721681860787 |
| Piecewise linear over 0.5, 4, 7, 14, 28 | 0.560990230501 | 0.364739525724 |

Kleyman *et al*. eLife 2017;6:e18541. DOI: 10.7554/eLife.18541

