## [Decision Letter]

Thank you for submitting your article "Determining sampling rates for high-throughput time series studies" for consideration by *eLife*. Your article has been reviewed by three peer reviewers, and the evaluation has been overseen by a Reviewing Editor and Aviv Regev as the Senior Editor. The reviewers have opted to remain anonymous.

The reviewers have discussed the reviews with one another and the Reviewing Editor has drafted these comments in an effort to crystallize our concerns about the novelty of this work. Before we render a binding decision, the Board asks that you respond soon to the essential concerns below in order to help us assess the technical advance you have achieved.

Summary:

Time course studies are critical for understanding biological processes. Facilitated by advances in sequencing technologies, high-throughput time series analyses are becoming more and more prevalent, and often more than one type of data is collected for each time point in order to probe/understand various aspects of cellular activity. Selecting the precise time points to use in such studies is challenging and often ad-hoc. In this manuscript, Kleyman et al. describe Time Point Selection (TPS), a method designed to address the problem of determining sampling rates in time series studies. The idea behind TPS is as follows:

1) start with a small set of genes known to be important for the process to be studied;

2) use cheap array-based experiments (e.g. NanoString) to measure expression of these genes at a high, uniform, sampling rate;

3) use the data to infer a subset of points that can be used to optimally "reconstruct" the entire data set from step 2 (importantly, the authors implement a rigorous, spline-based procedure for identifying optimal time points);

4) use the time points selected in step 3 for the genome-wide experiments.

The authors applied TPS to lung development in mice, and showed that the time points identified by TPS are better than uniform sampling and even phenotype-based sampling (in terms of reconstructing the mRNA profiles of the selected genes), and can be used not only for time series expression data, but also for miRNA and methylation time series data. The general strategy implemented in TPS has potential and could prove useful for future time course studies.

Essential concerns:

1) All the reviewers felt that the method is only compared to the most trivial baselines (uniform and random). E.g. there is no comparison to even the authors’ own 2005 paper on a closely related setup. There are various existing works on selecting knots for splines. One could use them, for example, by first fitting a spline to all the high-rate data, and then select points for that curve. Does TPS work better than non-trivial baselines? The authors must perform systematic comparisons to related work such as Rosa et al. 2012, Optimal Timepoint Sampling in High-Throughput Gene Expression Experiments. Further, the paragraph in the Discussion section showing the advantage of using TPS over phenotype-based sampling should be moved into the main text and presented clearly as this is a critical part of the paper.

2) All the reviewers also felt the manuscript required additional clarity and specific comments are below.

3) Since TPS uses an iterative algorithm for solving an optimization problem with many local optima, the initialization procedure is critical. I appreciate the fact the authors tested multiple procedures for initialization, and when using TPS one can try all initialization procedures (and in the end select the one with the best results on the left-out data). It is not clear, however, whether the software implemented by the authors tests all these initialization procedures.

4) In addition, the initialization metrics/methods are confusing. The Methods and Supplementary Methods sections talk about "uniform", "intervals of equal sizes", "max distance", metricA, metricB, and metricC. But it is not clear what initialization method was used in different analyses/figures. What procedure was used in Figure 1? The figure caption should mention that. The authors should decide on specific names for the different initialization methods, and use them consistently throughout the paper. Also, some figures present redundant results. Figure 1—figure supplement 1 presents results for "max distance", metricA, metricB, and metricC. Figure 1—figure supplement 2 presents "max distance", metricA, metricB, and metric, random points. So Figure 1—figure supplement 1 could be removed.

5) The "iterative improvement" procedures are also confusing. The method presented in Methods optimizes the mean squared error (on the left out data points) by iteratively removing/adding one point at a time. I assume this method was used in Figure 1. Then, in Figure 2, what does "absolute difference mean"? (I assume this refers to an initialization procedure, right?) The way the authors name the combinations of search and initialization procedures in Figure 2 is very confusing. Some names reflect an initialization procedure (Uniform Selection), other a search procedure (Simulated Annealing), others an evaluation criterion (Weighted Error). Thus, the results presented in Figure 2 are confusing, and neither the caption nor the main text clarify what is being tested. Similarly for Figure 4.

6) For the miRNA data, the authors perform a comparison of results using mRNA versus miRNA data for deciding the sampling time points. But the way the results are presented is again confusing. In the main text the authors say: "The results are presented in Figure 4. In addition to the comparison included in the mRNA figure, the miRNA figure includes the optimal results for using the actual miRNA data (as opposed to mRNA data) to select the points." But the results using the miRNA data are *not* in that figure.

7) It is nice that the project has a website but it looks mostly incomplete. Can the authors make the site a full-fledged web service so novice users and biologists with no programming background can use the software? Users should be able to upload a time series dataset and receive results with interactive visualization.

8) There is only one case study. The method is described as a broad method, but since it was applied to one process, it should be stated more carefully that it is a general method. Perhaps the authors should focus on this case study without the need to "sell" the method as the primary focus of the study.

9) The examples in Figure 6 are impressive. But how typical are they? What is the distribution of MSE across genes with TPS and with predefined timepoints?

10) Performance curves (Figure 2, Figure 4). Is the benefit due to a small number of genes, or consistent across the genome? It would be useful to show scatter plot (across genes) of MSE with TPS and with the best baseline.

11) Figure 5. How were these three genes selected? Are the correlations surprisingly strong when corrected for multiple comparisons? In 5A, the correlation is probably dominated by an outlier (third green marker) otherwise It is hard to see that the two curves are strongly anti-correlated. A similar outlier is also in Figure 6—figure supplement 1. Accompanying with a scatter plot would clarify the relation between the two. Also, compute confidence-interval for that correlation using bootstrap.

12) Figure 2: Seems that the uniform baseline was omitted from the left panel, please add it. Please also provide error bars for TPS and uniform baseline (on the left panel) by bootstrapping across the set of genes.

13) Subsection “Identified time points using mRNA data are appropriate for miRNA profiling”, second paragraph: "Using the selected based on mRNA data […] random points (p<0.01)". What is the p-value for the uniform baseline?

14) There is a line in the paper that says 'performance is very similar [...].0.43 […] 0.40". Provide a measure supported by a statistical test.

[Editors' note: further revisions were requested prior to acceptance, as described below.]

Thank you for resubmitting your work entitled "Selecting the most appropriate time points to profile in high-throughput studies" for further consideration at *eLife*. Your revised article has been favorably evaluated by Aviv Regev as the Senior Editor, a Reviewing Editor, and two reviewers.

The manuscript has been improved but there are some remaining issues that need to be addressed before acceptance, as outlined below by the referees. We ask that you pay close attention to these specific items, which we call out here, and are in the specific reviews below:

1) The revised version still has formatting issues. The response refers to Table 5, but no such table is given; figures are not numbered and provided separately from their captions, which makes it hard to follow and comment.

2) The new Figure 5 replaced outlier genes. The main text should explain how the 3 examples shown were selected.

3) We do not see a benefit in having the left panel of Figure 2. In their response, the authors say they wanted to show the difference between TPS and uniform selection. But panel (A) shows random selection, not uniform selection. Random selection is a made-up baseline designed to make TPS look good. Now that they have better baselines comparisons, the paper does not need it, and panel (A) can be removed.

4) Figure 3. Replace "Rem points" with "remaining points".

5) The authors mentioned that they "improved the caption of Figure 1 to explicitly state which initialization method is used for each result". But Figure 1 caption is unchanged. (They did clarify other figure captions.)

6) The authors added Figure 4—figure supplement 3 to show the performance of TPS trained on miRNA data. But that plot could/should be included directly in Figure 4. Otherwise it is hard for the reader to compare TPS when trained on mRNA versus miRNA data. Also, since there are several versions of TPS, which one was used in Figure 4—figure supplement 3? These kinds of details are still missing.

7) Some of the figures are still of inadequate quality. Some plots have a strange aspect ratio (e.g. Figure 5). Several plots do not have a y-axis label (e.g. Figure 3). Sometimes the labels are incomplete (e.g. "Error" in the comparison to Singh et al.; is it mean squared error?) Overall, the paper is still difficult to follow. (The fact that all references appear as "?" in the PDF also makes the paper hard to read. I assume the text file was not processed correctly during submission. But shouldn't the author verify that before submitting the paper?)

*Reviewer #1:*

The authors addressed most of the concerns raised. Most importantly, the authors added some evaluation to other approaches for selecting time points, and replaced results based on outliers.

The revised version still has formatting issues. The response refers to Table 5, but no such table is given; figures are not numbered and provided separately from their captions, which makes it hard to follow and comment.

*Reviewer #2:*

The authors addressed several comments from reviewers. I am now ok with the substance of the paper. While it would be useful to see validation on additional data/systems, I appreciate the difficulties in getting additional data sets for further testing of the method.

---

## [Author Response]

*Essential concerns:*

*1) All the reviewers felt that the method is only compared to the most trivial baselines (uniform and random). E.g. there is no comparison to even the authors’ own 2005 paper on a closely related setup. There are various existing works on selecting knots for splines. One could use them, for example, by first fitting a spline to all the high-rate data, and then select points for that curve. Does TPS work better than non-trivial baselines? The authors must perform systematic comparisons to related work such as Rosa et al. 2012, Optimal Timepoint Sampling in High-Throughput Gene Expression Experiments. Further, the paragraph in the Discussion section showing the advantage of using TPS over phenotype-based sampling should be moved into the main text and presented clearly as this is a critical part of the paper.*

The reviewers are correct that two prior studies (one from our group) have discussed theoretical ideas about determining which points to sample. However, there are a number of key differences between TPS and these prior studies. The most important difference is the fact that both prior methods mentioned above require an iterative process. These methods start by profiling all genes at a small number of time points. Based on these initial experiments they select another time point, profile it etc. until they reach some stopping criteria. This strategy, while theoretically interesting, is not practical (and indeed, to best of our knowledge have not been used so far). The main issue is the fact that such a strategy can take a very long time to complete. Given sample preparations which should be performed several times (at each iteration) and the sequencing itself, which has to be done one at a time, the entire process of selecting the time points can take weeks and even month for an experiment with 20 total time points such as the one we discuss. It is very unlikely that anyone would be willing to spend this much time. In addition, such iterative process introduces several new issues including the fact that each time point is prepared and sequenced on a different day, which has been shown to introduce biases and the fact that there is no real way to tell when to stop. Unlike TPS that starts with oversampling, and can thus determine accuracies for the subset of selected points, the iterative methods cannot compare their final decisions to any true profile and so are much more likely to reach a local minimum. Finally, the Rosa et al. method mentioned above relies on the availability of accuracy of related gene expression experiments. While this may be O.K. for some biological processes that have already been studied, when studying a new process or a new treatment this method cannot be applied since it is likely that no such relevant datasets exist and even if they do exist they may themselves be sampled at the wrong time points and thus no ‘ground truth’ exists for these methods. In contrast, TPS is both practical (requires only additional step), cheap and does not rely on the availability of other high throughput studies for the same system.

The above discussion refers to convenience and practicality. Given the above comment we also compared the accuracy of TPS to these prior methods. The Rosa et al. implementation did not work and even after consulting with the authors we were unable to use it. We were able to run the Singh et al. method and perform the comparison with that method. As can be seen in Figure 1—figure supplement 1, for all selections of a specific number of time points TPS outperforms the Singh et al. method, in some cases leading to over 40% reduction in reconstruction error.

We have revised the Introduction to discuss these prior methods and the difference between TPS and these methods and revise the supplement to discuss the comparison between the two methods.

Based on these comments we have modified the text. First, we moved the paragraph mentioned in the comment from Discussion to Results. We also added the following to Introduction “Relatively little work has focused so far on the selection of time points to sample in high throughput time series studies. […] In addition, these methods employ a stopping criteria that does not take into account the full profile and the Rosa et al. method also requires that related time series expression experiments be used to select the point, which may be problematic when studying new processes or treatments.”

*2) All the reviewers also felt the manuscript required additional clarity and specific comments are below.*

*3) Since TPS uses an iterative algorithm for solving an optimization problem with many local optima, the initialization procedure is critical. I appreciate the fact the authors tested multiple procedures for initialization, and when using TPS one can try all initialization procedures (and in the end select the one with the best results on the left-out data). It is not clear, however, whether the software implemented by the authors tests all these initialization procedures.*

We have implemented and made available all initialization methods in the software. As we now note in Discussion, since we are optimizing a specific function (MSE for selected genes at the selected time points) all initialization methods can be compared and the solution that leads to the lowest error be used. Thus, the large number of initialization methods should not be a problem for users of the method.

*4) In addition, the initialization metrics/methods are confusing. The Methods and Supplementary Methods sections talk about "uniform", "intervals of equal sizes", "max distance", metricA, metricB, and metricC. But it is not clear what initialization method was used in different analyses/figures. What procedure was used in Figure 1? The figure caption should mention that. The authors should decide on specific names for the different initialization methods, and use them consistently throughout the paper. Also, some figures present redundant results. Figure 1—figure supplement 1 presents results for "max distance", metricA, metricB, and metricC. Figure 1—figure supplement 2 presents "max distance", metricA, metricB, and metric, random points. So Figure 1—figure supplement 1 could be removed.*

As the reviewer suggested, we improved the caption of Figure 1 to explicitly state which initialization method is used for each result. We have also removed Figure 1—figure supplement 1 from the supplement.

*5) The "iterative improvement" procedures are also confusing. The method presented in Methods optimizes the mean squared error (on the left out data points) by iteratively removing/adding one point at a time. I assume this method was used in Figure 1. Then, in Figure 2, what does "absolute difference mean"? (I assume this refers to an initialization procedure, right?) The way the authors name the combinations of search and initialization procedures in Figure 2 is very confusing. Some names reflect an initialization procedure (Uniform Selection), other a search procedure (Simulated Annealing), others an evaluation criterion (Weighted Error). Thus, the results presented in Figure 2 are confusing, and neither the caption nor the main text clarify what is being tested. Similarly for Figure 4.*

We agree with the reviewer and now explicitly state, in the caption, which names correspond to different initializations for *our iterative method* and which are for baseline / comparison methods that are not iterative.

*6) For the miRNA data, the authors perform a comparison of results using mRNA versus miRNA data for deciding the sampling time points. But the way the results are presented is again confusing. In the main text the authors say: "The results are presented in Figure 4. In addition to the comparison included in the mRNA figure, the miRNA figure includes the optimal results for using the actual miRNA data (as opposed to mRNA data) to select the points." But the results using the miRNA data are not in that figure.*

The reviewer is correct that the results were not shown in the figure. We have now added a new figure with these results (Figure 4—figure supplement 3) and changed the text to: “Figure 4—figure supplement 3 presents the error achieved when using the miRNA data itself to select the set of points (evaluated on the miRNA data). […] For example, when using the 13 selected mRNA points, the average mean squared error is 0.4312 whereas when using the optimal points based on the miRNA data itself the error is 0.4042.”

*7) It is nice that the project has a website but it looks mostly incomplete. Can the authors make the site a full-fledged web service so novice users and biologists with no programming background can use the software? Users should be able to upload a time series dataset and receive results with interactive visualization.*

The software actually provides a graphical user interface which allows the user to determine how many time points to use by displaying different error levels for different selections and also the optimal time points for each selection. The software would be publicly available but we prefer not to implement it as a webserver. We had good success with prior stand-alone software tools we released (for example STEM and DREM) and the use of this one would also be easy and intuitive. We decided not to provide a webserver since some researchers prefer to not upload new data to public webservers and the ability to use the software on their own machines would be a plus for these individuals whereas it should not be detrimental to others.

*8) There is only one case study. The method is described as a broad method, but since it was applied to one process, it should be stated more carefully that it is a general method. Perhaps the authors should focus on this case study without the need to "sell" the method as the primary focus of the study.*

While it is indeed focused on one biological process, we believe the analysis is pretty comprehensive. Specifically, we tested it on several different types of high throughput biological data (admittedly for the same process, but still a much more rigorous analysis than prior papers mentioned by the reviewers above). Still, to address this issue we added the following to Discussion “We evaluated TPS on several types of high throughput data. However, we have only tested it so far on data for a specific biological process (lung development in mice). While we believe that such data is both challenging and representative and thus provides a good test case for the method, analysis of additional datasets may identify new challenges that we have not addressed and we leave it to future work to address these.”’

*9) The examples in Figure 6 are impressive. But how typical are they? What is the distribution of MSE across genes with TPS and with predefined timepoints?*

This comment is addressed by Appendix-Table 4 which shows the difference in error from different sampling rate methods (TPS and prior sampling rates). As can be seen, TPS does much better even when considering the MSE across all genes.

*10) Performance curves (Figure 2, Figure 4). Is the benefit due to a small number of genes, or consistent across the genome? It would be useful to show scatter plot (across genes) of MSE with TPS and with the best baseline.*

Again, as shown by the much better means, but also much better STD values (Appendix-Table 4), the performance of TPS is consistently better across most of the genes and is not a function of a few, very noisy, ones.

*11) Figure 5. How were these three genes selected? Are the correlations surprisingly strong when corrected for multiple comparisons? In 5A, the correlation is probably dominated by an outlier (third green marker) otherwise It is hard to see that the two curves are strongly anti-correlated. A similar outlier is also in Figure 6—figure supplement 1. Accompanying with a scatter plot would clarify the relation between the two. Also, compute confidence-interval for that correlation using bootstrap.*

Based on this comment we have performed the bootstrapping proposed by the reviewers and we now include the result as Figure 5—figure supplement 2. As can be seen, for most genes the observed correlation (based on all points) and the bootstrapped results are quite similar. However, for some genes there is indeed an outlier effect. Based on this result we now only include in Figure 5 genes that show consistent (negative) correlation in the bootstrap analysis. Note that time series methylation studies were only performed for 12 genes and comparison plots for *all* is available on the supporting website.

We also note that these correlation values are not a major aspect of the paper (the key for us is whether we can reconstruct the curves based on the time points selected) and were mainly presented to show that time series information could be useful to determine relationship between methylation and expression values.

*12) Figure 2: Seems that the uniform baseline was omitted from the left panel, please add it. Please also provide error bars for TPS and uniform baseline (on the left panel) by bootstrapping across the set of genes.*

The reason we removed the uniform sampling from 2B is for clarity. We wanted to have one clean figure showing the difference between the standard method (uniform) and the basic TPS method (with a uniform weight and greedy search). Then, after establishing the advantage of the general version of TPS in 2A, we show in 2B that we can further improve its results by improving the search method (either to avoid local minima using Simulated Annealing or when changing the uniform weighting). We now explain this in the caption.

*13) Subsection “Identified time points using mRNA data are appropriate for miRNA profiling”, second paragraph: "Using the selected based on mRNA data […] random points (p<0.01)". What is the p-value for the uniform baseline?*

The p-value is based on repeated (100) random samplings of time points and we have now clarified this in the text.

*14) There is a line in the paper that says 'performance is very similar [...].0.43 […] 0.40". Provide a measure supported by a statistical test.*

As discussed in the response to 13, we evaluated error significance using errors produced by random sets of points. For these we see standard deviations in the range of 0.05 as can be seen in Figure 4 but the reviewer is correct that ‘similar’ may be misplaced here since we do not provide any statistical guarantees. Instead, we have changed the text to say “As expected, the performance when using the miRNA data itself is better than when using the mRNA data. However, when taking into account the inherent noise in the data the differences are not large. For example, when using the 13 selected mRNA points, the average mean squared error is 0.4312 whereas when using the optimal points based on the miRNA data itself the error is 0.4042.”

[Editors' note: further revisions were requested prior to acceptance, as described below.]

*The manuscript has been improved but there are some remaining issues that need to be addressed before acceptance, as outlined below by the referees. We ask that you pay close attention to these specific items, which we call out here, and are in the specific reviews below:*

*1) The revised version still has formatting issues. The response refers to Table 5, but no such table is given; figures are not numbered and provided separately from their captions, which makes it hard to follow and comment.*

This is indeed our mistake. Table 5 should have been referring to Appendix-Table 4 and we have now fixed all references in the text to the correct table. As for the comment about the figures, this is because of the *eLife* upload instructions and policy. In the initial version we uploaded figures and captions together but were told by the journal editorial team to upload figures separately. We believe this would be fixed after acceptance.

*2) The new Figure 5 replaced outlier genes. The main text should explain how the 3 examples shown were selected.*

Indeed, in response to a comment in the initial round of review we replaced some of the genes presented in Figure 5. We now state that ‘These were the genes with the strongest negative correlation between their methylation and expression.’ And refer to Figure 5—figure supplement 2, which presents results for all genes in our study.

*3) We do not see a benefit in having the left panel of Figure 2. In their response, the authors say they wanted to show the difference between TPS and uniform selection. But panel (A) shows random selection, not uniform selection. Random selection is a made-up baseline designed to make TPS look good. Now that they have better baselines comparisons, the paper does not need it, and panel (A) can be removed.*

As the reviewer suggested we removed this panel to the supplement as ‘Figure 2—figure supplement 4’

*4) Figure 3. Replace "Rem points" with "remaining points".*

Fixed.

*5) The authors mentioned that they "improved the caption of Figure 1 to explicitly state which initialization method is used for each result". But Figure 1 caption is unchanged. (They did clarify other figure captions.)*

The reviewer is correct, we meant to say that we revised the captions for Figure 2 and Figure 4 as the reviewers requested in the previous round. These figures are the ones showing the results the reviewer refers to. Figure 1 is an overview illustrative figure and does not show any specific results. Thus, it does not use any initialization method and so no additions were needed for its caption.

*6) The authors added Figure 4—figure supplement 3 to show the performance of TPS trained on miRNA data. But that plot could/should be included directly in Figure 4. Otherwise it is hard for the reader to compare TPS when trained on mRNA versus miRNA data. Also, since there are several versions of TPS, which one was used in Figure 4—figure supplement 3? These kinds of details are still missing.*

As the reviewer suggested we now added Figure 4—figure supplement 3 as Figure 4 and we now explicitly mention in the caption that the max absolute difference initialization was used for these results.

*7) Some of the figures are still of inadequate quality. Some plots have a strange aspect ratio (e.g. Figure 5). Several plots do not have a y-axis label (e.g. Figure 3). Sometimes the labels are incomplete (e.g. "Error" in the comparison to Singh et al.; is it mean squared error?) Overall, the paper is still difficult to follow. (The fact that all references appear as "?" in the PDF also makes the paper hard to read. I assume the text file was not processed correctly during submission. But shouldn't the author verify that before submitting the paper?)*

The reference issue was fixed earlier and resulted from problems related to the *eLife* conversion of the latex file. As for the figures, we added y-axis labels of relative expression to Figure 6, changed the y-axis label from error to mean squared error for Figure 1—figure supplement 1, and rescaled Figure 5 so the aspect ratio looked better. We made similar changes to all relevant supplementary figures as well.